# Faster approximate subgraph counts with privacy

**Dung Nguyen**
Department of Computer Science and
UVA Biocomplexity Institute and Initiative
University of Virginia, USA
`dungn@virginia.edu`

**Mahantesh Halappanavar**
Data Sciences and Machine Intelligence Group
Pacific Northwest National Laboratory, USA
`hala@pnnl.gov`

**Venkatesh Srinivasan**
Department of Mathematics and
Computer Science
Santa Clara University, USA
`vsrinivasan4@scu.edu`

**Anil Vullikanti**
Department of Computer Science and
UVA Biocomplexity Institute and Initiative
University of Virginia, USA
`vsakumar@virginia.edu`

## Abstract

One of the most common problems studied in the context of differential privacy for graph data is counting the number of non-induced embeddings of a subgraph in a given graph. These counts have very high global sensitivity. Therefore, adding noise based on powerful alternative techniques, such as smooth sensitivity and higher-order local sensitivity have been shown to give significantly better accuracy. However, all these alternatives to global sensitivity become computationally very expensive, and to date efficient polynomial time algorithms are known only for few selected subgraphs, such as triangles, $k$-triangles, and $k$-stars. In this paper, we show that good approximations to these sensitivity metrics can be still used to get private algorithms. Using this approach, we much faster algorithms for privately counting the number of triangles in real-world social networks, which can be easily parallelized. We also give a private polynomial time algorithm for counting any constant size subgraph using less noise than the global sensitivity; we show this can be improved significantly for counting paths in special classes of graphs.

## 1   Introduction

The notion of Differential Privacy (DP) [12] has emerged as the *de facto* standard notion for supporting queries on private and sensitive data. DP ensures that changes in private data have limited statistical influence (measured by the privacy budget $\epsilon, \delta$) on the output of queries, without needing any attack model, which is one of the reasons of its popularity. Networked abstractions are commonly used in a number of applications, such as public health, social networks, and finance. Such datasets are usually sensitive, and maintaining privacy is an important concern. Within the context of network/graph data, two privacy models have been considered in most prior work, namely, edge and node privacy. There has been a lot of interest in developing private algorithms for a number of network problems, such as community detection [29, 9, 8, 22], and counting small subgraphs (e.g., stars and triangles [25, 36, 23, 31]).

However, efficient private algorithms with good accuracy are not known for many graph problems, including counting small subgraphs. One of the main challenges is that graph problems often have very high *global sensitivity* (the maximum change in the output function due to the change in one edge/vertex (see Section 3)), in contrast to many statistical and machine learning queries. As a result, DP algorithms based on global sensitivity (which can usually be computed quite easily for many problems) do not give good accuracy bounds, in general. For instance, in the edge-privacy

37th Conference on Neural Information Processing Systems (NeurIPS 2023).

model, the global sensitivity of #triangles in a graph $G = (V, E)$ can be as high as $n - 2$, where $n = |V|$, making the added noise too large in many instances. A number of novel alternatives to global sensitivity have been proposed, e.g., smooth, restricted and multi-level local sensitivity, and ladder functions [36, 23, 3, 31], which add significantly lower level of noise than mechanisms based on global sensitivity.

However, all these alternative notions of sensitivity are computationally much more challenging, and none of the current graph DP algorithms for subgraph counting scale to even moderate size networks. Even the simplest problem of counting #triangles privately takes $\min\{m \max_v d(v), M(n)\}$ time [23], where $m = |E|$, $d(v)$ is the degree of node $v \in V$, and $M(n)$ is the time for multiplying two $n \times n$ matrices, which is super-quadratic, in general. This is a sharp contrast with the significant advances in non-private graph mining where complex network properties such as clique counts can be computed using provably-efficient and practical algorithms [35, 20, 19]. For a few other subgraphs, such as $k$-cliques and $k$-triangles (see [36, 23] for definitions), private algorithms using noise lower than the global sensitivity are known, but the worst case running time can still be $O(n^k)$ [36]. However, for most subgraph counting problems, *no private algorithms which use alternatives to global sensitivity, and have time comparable to the non-private algorithms, are known*.

A main tool for subgraph counting and other graph mining tasks in massive networks are sampling based approximation algorithms, e.g., [2]; however, these haven't been used in the context of private graph algorithms. In this paper, we take the first steps in this direction. Our main contributions are:

● We show that for a query $f(G)$, we can get a private algorithm by adding noise that scales with an approximation to the smooth sensitivity to an approximation to $f(G)$ (Theorem 1). This result opens the door to improving efficiency of private algorithms using approximation techniques that have been very useful in non-private graph analytics. As an illustration, we use this result to obtain a quasilinear time private algorithm for counting #triangles under edge privacy (Theorem 2) for graphs which satisfy a stronger transitivity property (see Definition 5). Our algorithm adapts the *diamond sampling* technique of [2] for approximate triangle counting, and can also be parallelized easily. No parallel algorithms have been developed for private subgraph counting so far.

● Extending the approach of [23], we design a private higher-order local sensitivity approach for subgraph counting, which gives privacy even when the higher-order local sensitivities are estimated approximately (Theorem 7). This yields the first private algorithm to count the number of embeddings of *any* subgraph with $\ell$ edges in time $O(n^{2\ell})$, using less noise than global sensitivity. For the specific case of paths with $\ell$ edges, we develop an algorithm that gives a very close approximation to the higher-order sensitivity in time $O(h(d, t, \ell)poly(n))$, where $d$ and $t$ are the degeneracy and treewidth of $G$, respectively (defined later). Thus, for paths, the running time does not grow as $O(n^\ell)$ for graphs with bounded degeneracy and treewidth. These properties have been used extensively in developing efficient algorithms for subgraph counting and other graph mining tasks, e.g., [5, 13, 6]; but, they haven't been used for private algorithms.

● We experimentally evaluate our algorithm for counting #triangles (Section 6). We show that our algorithm for estimating the smooth sensitivity has very good approximation factor, and significantly better running time than the exact algorithm. Using approximate smooth sensitivity for counting #triangles gives good accuracy even for fairly low $\epsilon$ values, suggesting that using approximate smooth sensitivity does not compromise performance. Our algorithm also has good scaling, and is the first private algorithm for counting #triangles which has been run on networks with over 2 million edges.

We note that many details, including proofs are presented in the Supplementary Material (SM).

## 2   Related Work

The area of DP graph algorithms is quite large and active. There has been a lot of work on DP algorithms for many basic graph problems, such as community detection, subgraph counting, finding small cuts, and releasing synthetic graphs [25, 28, 30, 33, 16, 3, 15, 21, 17]. There is some work on graph algorithms under local DP, e.g., [16], but most of the prior work has been for global DP (as defined in Section 3), and is our focus in this work. We refer to [27] for a recent survey on private graph algorithms. For brevity, we only discuss prior work on private subgraph counting that is directly related to our paper, and in the edge-DP model. We omit the discussion on private estimation of edge

| | Privacy model | Runtime |
|---|---|---|
| Global Sensitivity | $\epsilon$ | $O(1)$ |
| Ladder function [10] | $(\epsilon, \delta)$ | $O(\text{matrix mul. of size } n)$ |
| Recursive mechanism [7] | $(\epsilon, \delta)$ | $O(mn)$ |
| Restricted Sensitivity [3] | $(\epsilon, \delta)$ | $O(mn)$ |
| Blackbox Transformation [4] | $(\epsilon, \delta)$ | $O(\text{non-private } \#_\Delta \text{ alg.})$ |
| (Exact) Smooth Sensitivity [23] | $(\epsilon, \delta)$ | $O(\text{matrix mul. of size } n)$ |
| **Our method** | $(\epsilon, \delta)$ | $O(m \log^2 n)$ for $(C, \Lambda)$-graphs with constant $C, \Lambda$ |

Table 1: Summary of the characteristics of differentially private #triangle counting algorithms in the edge-privacy model (the other works mentioned here consider other subgraphs also, but we only focus on the runtime of counting #triangles). The runtime of Blackbox Transformation [4] includes the calculation of an approximated triangle counts, while all others assume the true counts are given. The runtime of our method is reported with $\delta = \Omega(n^{-2})$ and constant $\gamma$ (see Corollary 1).

density and degree distribution, which has been studied more extensively, e.g., [14], and results for node-DP [25], which is less studied.

As mentioned earlier, global sensitivity, denoted by $GS(f)$ for a graph metric $f(G)$ (defined in Section 3) is high for many subgraph counting problems. Adding noise based on $GS(f)$ leads to low accuracy, while noise based on just $LS_f$, the local sensitivity, need not be private [31]. Nissim et al. [31] develop the notion of smooth sensitivity $S^*_{f,\beta}(G) = \max_{G'} \left( LS_f(G') \cdot e^{-\beta \cdot d(G,G')} \right)$, where $d(G, G')$ is the swap distance between graphs $G$ and $G'$, and show that adding noise based on $S^*_{f,\beta}(G)$ is private. However, smooth sensitivity becomes computationally much more challenging, since its definition involves considering local sensitivity at distance $t$ for all $t$, which doesn't seem like a polynomial time computation. Nissim et al. [31] develop polynomial time algorithms for computing the smooth sensitivity exactly for counting #triangles, which was improved slightly to $\min\{md_{max}, M(n)\}$ [23], where $M(n)$ is the matrix multiplication time. Polynomial time smooth sensitivity bounds were also shown for a small number of other subgraphs, but no polynomial time algorithms are known beyond that. Interestingly, no hardness bounds are known for smooth sensitivity; the existing hardness bounds, e.g., [23, 36] only give hardness for computing local sensitivity at distance $t$. In order to handle other subgraph counts, Karwa et al. [23] develop a different technique involving local sensitivity of local sensitivity (motivated by the propose-test-release technique [12]), and use it for private counts of $k$-cliques. Zhang et al. [36] develop the technique of *ladder functions* to handle other kinds of subgraphs, and use it to privately count #$k$-cliques in the graph in time $O(nT(n))$, where $T(n)$ is the time needed to count #$k$-cliques non-privately. A few other techniques, such as inverse sensitivity [1] and propose-test-release [12] are known. However, private algorithms for counting most subgraphs including paths and trees are not known. The recent work of [4] is related, but only considers approximation in the queries, but not in sensitivity. As a result, our methods give significantly higher efficiency. We also note that Blocki et al. [4] develop a black-box approach to make certain approximation algorithms differentially private. However, their work requires the function being computed to have "small" global sensitivity.

The difficulty of subgraph counting with DP has motivated work in slight variations of the DP model. Rastogi et al. [34] consider the problems of releasing more general subgraph counts. However, they consider a relaxed version of edge-DP, called (edge) adversarial privacy, that uses a Bayesian attacker. Chen et al. [7] design a different approach that gives lower bounds for general subgraph counts through a linear program to form a recursive strategy. But, as mentioned in [36], their method suffers from a bias between the true query answer and the lower bound, in exchange for less noise. [32] presents a different approach based on iterative refinement that estimates counts by degree, and can be implemented in time $O((\max_v deg(v))^3 m)$.

There have been studies on subgraph counting (including triangle counting) under other privacy models: the node-DP model [25, 10], and the shuffle model [18]. We note that our analysis of approximate smooth sensitivity holds for other privacy models as well, and we expect this could be used for improved private queries for other problems. However, the specific technique using diamond sampling for faster triangle counting only holds in the edge-DP model. Fundamentally new ideas are needed for extending this technique to node-DP and others. For a more comprehensive review of recent development on subgraph counting with privacy, we refer readers to [27].

# 3 Preliminaries

A non-induced embedding of a graph $H = (V_H, E_H)$ into a graph $G = (V, E)$ is a mapping $\phi : V_H \to V$ such that $(\phi(u), \phi(v)) \in E$ whenever $(u, v) \in E_H$. We use $f_H(G)$ to denote the number of non-induced embeddings of a $H$ in $G$. We drop the subscript $H$ when it is clear from the context. Let $\ell = |V_H|$ and $n = |V|$. $A$ denotes the adjacency matrix of graph $G$. Let $N(i)$ and $\overline{N(i)}$ denote the set of neighbors and non-neighbors of node $i \in V$, respectively; let $d(i) = |N(i)|$ and $\bar{d}(i) = n - 1 - d(i)$. Let $d_{max} = \max_{v \in V} d(v)$ denote the maximum degree.

**Differential privacy on graphs.** Let $\mathcal{G}$ denote a set of graphs on a fixed set $V$ of nodes. For a graph $G \in \mathcal{G}$, we use $V(G)$ and $E(G)$ to denote the set of nodes and edges of $G$, respectively. In this paper, we will focus on the notion of *edge privacy* [3], where all graphs $G \in \mathcal{G}$ have a fixed set of nodes $V(G) = V$, and two graphs $G, G' \in \mathcal{G}$ are considered neighbors, i.e., $G \sim G'$, if they differ in exactly one edge, i.e., $|E(G) - E(G')| = 1$.

**Definition 1.** *A (randomized) algorithm $M : \mathcal{G} \to R$ is $(\epsilon, \delta)$-differentially private if for all subsets $S \subset R$ of its output space, and for all $G, G' \in \mathcal{G}$, with $G \sim G'$, we have $Pr[M(G) \in S] \leq e^\epsilon Pr[M(G') \in S] + \delta$ [12, 3].*

**Problem statement: subgraph counting with edge differential privacy.** Given a family of graphs $\mathcal{G}$ on a set $V$ of vertices, a subgraph $H$, and parameters $\epsilon, \delta$, construct an $(\epsilon, \delta)$-differentially private mechanism $M_{f_H} : \mathcal{G} \to 2^V$, such that $|M_{f_H}(G) - f_H(G)|$ is minimized.

We discuss the notions of sensitivity mostly using the notation from [23, 31], with slight changes. Let $LS_f(G) = \max_{G' \sim G} |f(G) - f(G')|$ denote the local sensitivity of $f$; we also use $LS(f(G))$ to denote this. $GS(f) = \max_G LS(f(G))$ denotes the global sensitivity. The local sensitivity of a function $f$ on a graph $G$ at distance $t$ is defined as $LS_f^{(t)}(G) = \max_{d(G,G') \leq t} LS_f(G')$

**Definition 2.** *(Smooth bound on LS [31]) For $\beta > 0$, a function $S : D^n \to \mathbb{R}^+$ is a $\beta$-smooth upper bound on $LS_f$ if it satisfies the following conditions: (1) for all $x \in D^n$: $S(x) \geq LS_f(x)$, and (2) for all $x \sim y \in D^n$: $S(x) \leq e^\beta \cdot S(y)$.*

$S_{f,\beta}^*(G) = \max_{G'}(LS(f(G')) \cdot e^{-\beta d(G,G')})$ is the smallest function satisfying Definition 2, and is referred to as the $\beta$-smooth sensitivity of $f$ at $G$. Using the local sensitivity at distance $t$, the smooth sensitivity can be written as $S_{f,\beta}^*(G) = \max_{t=1,\ldots,\binom{n}{2}} e^{-t\beta} LS_f^{(t)}(G)$.

**Definition 3.** *(Admissible Noise Distribution) [31] A probability distribution on $\mathbb{R}^d$ given by a density function $h$ is $(\alpha, \beta)$-admissible if, for $\alpha = \alpha(\epsilon, \delta)$, $\beta = \beta(\epsilon, \delta)$, the following conditions hold for all $\Delta \in \mathbb{R}^d$ and $\lambda \in \mathbb{R}$ satisfying $||\Delta||_1 \leq \alpha$ and $|\lambda| \leq \beta$, and for all measurable $\mathcal{S} \subseteq \mathbb{R}^d$: (1) Sliding property: $\Pr_{Z \sim h}[Z \in \mathcal{S}] \leq e^{\epsilon/2} \Pr_{Z \sim h}[Z \in \mathcal{S} + \Delta] + \delta/2$, and (2) Dilation property: $\Pr_{Z \sim h}[Z \in \mathcal{S}] \leq e^{\epsilon/2} \Pr_{Z \sim h}[Z \in e^\lambda \cdot \mathcal{S}] + \delta/2$.*

An important example of an admissible noise distribution is the Laplace mechanism $Lap(\lambda)$ probability density function is $h(z) = \frac{1}{2\lambda} e^{-|z|/\lambda}$.

**Lemma 1.** *(Calibrating noise to Smooth Sensitivity [31]) Let $Z$ be a random variable sampled from an $(\alpha, \beta)$-admissible noise. Let $S_{f,\beta}$ be the $\beta$-smooth upper bound on the local sensitivity of $f$. Then algorithm $A(x) = f(x) + \frac{S_{f,\beta}(x)}{\alpha} Z$ is $(\epsilon, \delta)$-differentially private.*

**Definition 4.** *(An $(\alpha, \delta)$-approximation to a function $f$). $\tilde{f}$ is said to be an $(\alpha, \delta)$-approximation to a function $f$ if for any input $x$, with probability at least $1 - \delta$, we have $(1 - \alpha)f(x) \leq \tilde{f}(x) \leq (1 + \alpha)f(x)$.*

Social networks generally have the property that nodes have high clustering coefficient, which is the fraction of pairs of neighbors of a node which are connected. We consider a more restricted notion here. We refer to a path with $r$ edges as an $r$-path. We say a 2-path $i, j, k$ is "closed" if $(i, k) \in E$, i.e., $i, j, k$ form a triangle in $G$.

**Definition 5.** *($(C, \Lambda)$-transitive graph) A graph is said to be $(C, \Lambda)$-transitive if for any vertex $j \in V$ and edge $(i, j) \in E$, we have: if $\deg(j) > \Lambda$, then $C$ fraction of all the 2-paths starting with $i, j$ are closed.*

# 4 Approximate smooth sensitivity and fast private triangle counting

We first show that we can ensure privacy even if the smooth sensitivity $S^*_{f,\beta}$ is estimated approximately; we then extend it to show that this works when $f(G)$ and $S^*_{f,\beta}$ both are estimated approximately.

**Definition 6.** $\tilde{S}_{f,\beta}$ *is said to be a $\gamma$-upper approximation of the smooth sensitivity $S^*_{f,\beta}$ of function $f$ if $S^*_{f,\beta}(D) \leq \tilde{S}_{f,\beta}(D) \leq e^\gamma S^*_{f,\beta}(D)$ for any dataset $D$. $\tilde{S}_{f,\beta}$ is said to be a $(\gamma, \delta')$-upper approximation of the smooth sensitivity $S^*_{f,\beta}$ of function $f$ if $\tilde{S}_{f,\beta}$ is a $\gamma$-upper approximation for any dataset $D$ with probability at least $1 - \delta'$.*

We observe below that calibrating noise using a $(\gamma, \delta')$-approximation of the smooth sensitivity gives us privacy. We assume $f$ is a real valued function, since we are focused on graph statistics.

**Lemma 2.** *(Lemma 12) Let $\tilde{S}_{f,\beta}$ be a $(\gamma, \delta')$-approximation to $S^*_{f,\beta}$, for $\gamma = \beta = \frac{16\epsilon}{\ln(2/\delta)}$. Then,*

$$A(D) = f(D) + Lap(\frac{2\tilde{S}_{f,\beta}(D)}{\epsilon}) \text{ is } (\epsilon, \frac{e^{\epsilon/2}+1}{2}\delta + 2\delta')\text{-differentially private.}$$

**Approximate Smooth Sensitivity for Approximate Query.** We now show that approximate smooth sensitivity can be used even when $f(G)$ is computed approximately, which becomes a bit more challenging.

Let $A_f$ be $(\alpha_1, \delta_1)$-approximation of a function (query) $f$ for a small constant $\alpha_1 < 1/2$. Let $\tilde{S}_{f,\beta}$ be $(\gamma, \delta_2)$-upper approximation to $S^*_{f,\beta}$. We will show that we can utilize $A_f$ and $\tilde{S}_{f,\beta}$ to calculate a differentially private version of the function $f$.

For the purpose of privacy analysis, we define the functions $g_f$, $s_f$, and $S_{g_f}$ as below. Those functions may not be computed efficiently. However, they are only used for the analysis, and the actual algorithm does not compute them. The actual computation (Algorithm 3) will only utilize $A_f$ and $\tilde{S}_{f,\beta}$ to output the differentially private version of $f$. We follow some of the analysis of [4] to prove our privacy guarantees below. We share (with [4]) the same process of proving that $S_{g_f}(D)$ is a smooth-upper bound of the local sensitivity (Smooth Sensitivity) of $g_f$. The main difference is our analyses has to take into account the newly defined function $s_f$–which is a bounded variant of the Approximate Smooth Sensitivity ($\tilde{S}_{f,\beta}$, which approximates $S^*_{f,\beta}$), while [4] uses the global sensitivity $GS_f$ of $f$. As we can see in Lemma 13, the second condition requires $S_{g_f}(D) \leq e^{\beta'} S_{g_f}(D')$ for any pair of neighbor datasets $D \sim D'$. While $GS_f$ remains unchanged in both $D$ and $D'$, $\tilde{S}_{f,\beta}$ may have different values for $D$ and $D'$ which makes it more difficult to analyze $S_{g_f}$. We also have to take into account the small probability $\delta_2$ that $\tilde{S}_{f,\beta}(D) \notin \{S_{f,\beta}(D), e^\gamma S_{f,\beta}(D)\}$, that will add up in the $\delta$-part of $(\epsilon, \delta)$-DP of the final output.

**Definition 7.** *Let $A_f$ be an $(\alpha_1, \delta_1)$-approximation of $f$. We define functions $g_f$ and $s_f$ as:*

$$g_f = \begin{cases} A_f(D) \text{ if } (1-\alpha_1)f(D) \leq A_f(D) \leq (1+\alpha_1)f(D), \\ (1-\alpha_1)f(D) \text{ if } A_f(D) < (1-\alpha_1)f(D), \\ (1+\alpha_1)f(D) \text{ if } A_f(D) > (1+\alpha_1)f(D). \end{cases} \quad (1)$$

$$s_f = \begin{cases} \tilde{S}_{f,\beta}(D) \text{ if } S_{f,\beta}(D) \leq \tilde{S}_{f,\beta}(D) \leq e^\gamma S_{f,\beta}(D), \\ S_{f,\beta}(D) \text{ if } \tilde{S}_{f,\beta}(D) < S_{f,\beta}(D), \\ e^\gamma S_{f,\beta}(D) \text{ if } \tilde{S}_{f,\beta}(D) > e^\gamma S_{f,\beta}(D). \end{cases} \quad (2)$$

*Let $S_{g_f}(D) = 4\alpha_1 g_f(D) + 2s_f(D)$*

**Lemma 3.** *(Lemma 13) Given $g_f$, $s_f$, and $S_{g_f}$ as defined above, $S_{g_f}$ is a $\beta'$-smooth upper bound of the local sensitivity of $g_f$, where $\beta' \geq 4\alpha_1 + \gamma + \beta$ and $\alpha_1 < 1/2$.*

In order to prove that $S_{g_f}$ is a $\beta'$-smooth upper bound of the local sensitivity of $g_f$, we have to prove the two conditions: $LS_{g_f}(D) \leq Sg_f(D)$, and $S_{g_f}(D) \leq e^{\beta'} S_{g_f}(D')$ for any pair of neighbor datasets $D \sim D'$. The full proof is in Lemma 13 in the SM. Theorem 1 below summarizes our result.

**Theorem 1.** *(Theorem 4) Let $A_f$ be a $(\alpha_1, \delta_1)$-approximation to $f$, $\tilde{S}_{f,\beta}$ be a $(\gamma, \delta_2)$-upper approximation to $S^*_{f,\beta}$, for $\gamma = \beta = \frac{8\epsilon}{\ln(2/\delta)}$, $\alpha_1 = \frac{4\epsilon}{\ln(2/\delta)}$. Then, $A(D) = A_f(D) + Lap(\frac{8\alpha_1 A_f(D) + 4\tilde{S}_{f,\beta}(D)}{\epsilon})$ is $(\epsilon, \frac{e^{\epsilon/2}+1}{2}\delta + 2\delta_1 + 2\delta_2)$-differentially private.*

## 4.1 Application: fast private triangle counting

In this section, we study the problem of computing $f_\Delta(G)$, the number of triangles in a graph $G(E, V)$, by fast approximation of its smooth sensitivity. Recall the definitions of local sensitivity at distance $t$, denoted by $LS_f^{(t)}(G)$, defined in Section 3. We denote the local and smooth sensitivity of $f_\Delta(G)$ by $LS_\Delta^{(t)}(G)$ and $S_{\Delta,\beta}^*(G)$. Let $a_{ij} = \sum_{k \in [n]} A_{ik} A_{jk}$ denote the number of common neighbors of nodes $i$ and $j$ in a graph and $b_{ij} = \sum_{k \in [n]} A_{ik} \oplus A_{jk}$ denote the number of nodes that are neighbors of $i$ or $j$ but not both. Then it has been shown that $LS_\Delta^{(t)}(G) = \max_{i,j} c_{ij}(t)$ where $c_{ij}(t) = \min\left(a_{ij} + \frac{t+\min(t, b_{ij})}{2}, n-2\right)$ (Claim 3.13 of [31]). We can rewrite this as $LS_\Delta^{(t)}(G) = \min(\max_{i,j}\left(a_{ij} + \frac{b_{ij}}{2} + \frac{t}{2}\right), \max_{i,j} a_{ij} + t, n - 2) = \min(\max_{i,j} \frac{deg(i)+deg(j)}{2} + \frac{t}{2}, \max_{i,j} a_{ij} + t, n-2)$.

Figure 1: Paths $(i, k, j)$ and $(i, o, j)$ represent wedges formed by nodes $i, j$ [2].

The bottleneck in computing $LS_\Delta^{(t)}(G)$ (and $S_{\Delta,\beta}^*(G)$) is estimating $\hat{a} = \max_{i,j} a_{ij}$. We adapt the diamond sampling technique of [2] for this task. By doing that, we can quickly calculate $\widehat{LS}_\Delta^{(t)}(G) = \min(\max_{i,j} \frac{deg(i)+deg(j)}{2} + \frac{t}{2}, \hat{a} + t, n - 2)$ The core idea in diamond sampling is to find a "diamond" (a 4-cycle) of the form $(i, o, j, k)$ obtained by the intersection of two "wedges" (a 2-path) $(i, o, j)$ and $(i, k, j)$, as shown in Figure 1. Since any pair $(i, j)$ is part of $a_{ij}^2$ diamonds, the probability of selecting such a diamond is proportional to $a_{ij}^2$. This idea is formalized in Algorithm 4. Let $W$ be the matrix $W_{ki} = deg(k) \times deg(i)$ for all $(k, i) \in E$, and 0 for all other entries.

We first note that if $\hat{a} = 0$, it must be the case that $G$ has no paths of length $\geq 2$, for if there is a 2-path $i, k, j$, then $a_{ij} \geq 1$. Therefore, if $\hat{a} = 0$, we can determine $LS_\Delta^{(t)}(G)$ exactly in time $O(m)$. Therefore, we assume that $\hat{a} \geq 1$ in the rest of this section.

Even though Algorithm 4 can quickly estimate the quantity $\max a_{ij}^2$ within a specific bound, it can only do so with the number of iterations that is sufficiently large–which its exact threshold is unknown without the access to the dataset and the value of interest ($\max a_{ij}^2$) itself (see Lemma 15). We overcome this issue with the following idea. First, we propose some estimation $\tau$ of $\max a_{ij}^2$ and run an instance of Algorithm 4 with the number of iterations computed from $\tau$. Second, we compare the estimation of $\max a_{ij}^2$ returned by Algorithm 4 with $\tau$ to determine if we over- or under-estimate $\tau$. Third, we adjust $\tau$ accordingly and repeat the process until we find a good value of $\tau$.

---

**Algorithm 1** $\tau$ estimation
**Input:** $G(E, V)$
**Output:** $\tau : \tau < \max_{ij} a_{ij}^2$

1: $k := 0, \tau_k := n^2, \theta := \frac{1}{2}$
2: **while** TRUE **do**
3:     Calculate $s := \frac{3c \log n \|W\|}{\theta^2 \tau_k}$
4:     $x := Algorithm\ 4(G, s)$
5:     **if** $\max_{ij} x_{ij} \frac{\|W\|}{s} < \frac{3}{2}\tau_k$ **then**
6:       $\tau_{k+1} := \frac{3}{4}\tau_k$
7:       $k := k + 1$
8:     **else**
9:       Return $\tau_k$
10:    **end if**
11: **end while**

**Algorithm 2** Algorithm to compute $\tilde{S}_{\Delta,\beta}(G)$
**Input:** $G(E, V), \beta, \delta, \gamma$
**Output:** $\tilde{S}_{\Delta,\beta}(G)$

1: $\tau_k := Algorithm\ 1(G)$
2: $\tau := \tau_k/3$
3: $\theta := \frac{e^{2\gamma}-1}{e^{2\gamma}+1}$
4: $c := max(\log_n(1/(2\delta)) + 2, 12\theta^2)$
5: Calculate $s := \frac{3c \log n \|W\|}{\theta^2 \tau}$
6: $x := Algorithm\ 4(G, s)$
7: $\hat{a}^2 := \frac{\max_{ij} x_{ij} \frac{\|W\|}{s}}{1-\theta}$
8: Calculate $\widehat{LS}_\Delta^{(t)}(G), t = 1, \ldots, \log(n)/\beta$, using $\hat{a}$
9: Return $\max_{t=1,\ldots,\log(n)/\beta} e^{-t\beta} \widehat{LS}_\Delta^{(t)}(G)$

---

We implement Algorithm 1 based on this idea. In the algorithm, we first set $\tau$ to its largest possible value ($n^2$) knowing that we are over-estimating $\tau$ ($\tau > \max a_{ij}^2$). We note that by over-estimating $\tau$, Algorithm 4 (line 6) may not have enough iterations to guarantee the convergence of $\max x_{ij}$ to $\max a_{ij}^2$. Note that in the output of Algorithm 4, for any $i, j$, $x_{ij}$ is an approximation of $a_{ij}^2$. We

determine if we are truly over-estimating $\tau$ by applying Corollary 3 (in the SM), comparing the estimation $\max_{ij} x_{ij}$ from Algorithm 4 in line 6 to $(3/2)\tau$. The proof utilizes an idea that even though $\max x_{ij}$ may not converge to $\max a_{ij}^2$ and we may not determine it directly, $\max x_{ij}$ can converge to $\tau$ once we under-estimate $\tau$ (i.e., $\tau < \max a_{ij}^2$) and that we can check this convergence easily. Corollary 3 states that when $\max x_{ij} < (3/2)\tau$, $\max a_{ij}^2 < 3\tau$ which means we should lower $\tau$. We keep running Algorithm 4 and reducing $\tau$ by $3/4$ in each iteration until the estimation of $\max x_{ij}$ lies above $(3/2)\tau$. By Corollary 3, we show that when it happens, $\tau_k \le \max a_{ij}^2$ that guarantees the convergence of $\max x_{ij}$ to $\max a_{ij}^2$. Lemma 4 shows that when the algorithm outputs $\tau_k$, w.h.p., the true value of interest $\max a_{ij}^2$ is within $\tau_k$ and $4\tau_k$.

**Lemma 4.** *(Lemma 18 in SM) Let $\tau_k$ be the output of Algorithm 1. With probability at least $1 - 1/n^{c-3}$, $\tau_k < \max_{ij} a_{ij}^2 < 4\tau_k$.*

Algorithm 2 calculates an approximation of $\max a_{ij}^2$ for the Approximated Smooth Sensitivity with any $\gamma$ (which is required for Differential Privacy with an arbitrary $\epsilon$). Since $\tau_k$ is guaranteed to be close to the true value of $\max a_{ij}^2$ (within a constant factor of 4), we run the last instance of Algorithm 4 with some constant $\theta$, which is derived from the target approximation factor $\gamma$. Lemma 5 shows that the estimation output by Algorithm 4 in line 6 falls within the range of $(1 - \theta, 1 + \theta)$ of the true value. Following that, Theorem 2 guarantees that the quantity output by Algorithm 2 is a $(\gamma, \delta)$-upper approximation to Smooth Sensitivity $S_{\Delta,\beta}(G)$ of the triangle count of graph $G$.

**Lemma 5.** *(Lemma 19 in SM) With probability at least $1 - \delta$ in Algorithm 2, $\max_{ij} x_{ij} \frac{\|W\|}{s} \in (1 - \theta, 1 + \theta) \max_{ij} a_{ij}^2$.*

**Theorem 2.** *(Theorem 5, 6 in SM) Suppose $\max_{ij} a_{ij} > 0$. Algorithm 2 outputs $\tilde{S}_{\Delta,\beta}(G)$, which is a $(\gamma, \delta)$-upper approximation to $S_{\Delta,\beta}(G)$, in $O\left( \frac{\log_n\left(\frac{1}{2\delta}\right) \|W\|_1 \log m \log n}{\left(\frac{e^{2\gamma}-1}{e^{2\gamma}+1}\right)^2 \max_{ij} a_{ij}^2} + m + \frac{\log n}{\beta} + n \right)$ time.*

*Proof.* (Sketch) Algorithm 1 has a run-time of $kO(s' \log m))$ where $s' = O(\|W\|_1 c \log n/(\theta^2 \tau))$ as it executes Algorithm 4 with varying $s = s'$ and $k$ is the number of search steps. Since at the end of the search $\tau_k > \max_{ij} a_{ij}^2/4$ w.h.p., we know that $k = O(\log \frac{4n^2}{\max_{ij} a_{ij}^2})$. In the first step, we start with $\tau_0 = n^2$ and reduce $\tau_k$ by a factor of $3/4$ after each step. The total running time will be $O(\|W\|_1 \times \log m \times \frac{c \log n}{\theta^2} \times \frac{1}{n^2}(1 + (3/4)^{-1} + \ldots (3/4)^{-k}) = O\left( \frac{\|W\|_1 c \log m \log n}{\theta^2 \max_{ij} a_{ij}^2} \right)$. Finally, Algorithm 2 has a run-time of $O\left( \frac{\log_n (2\delta) \|W\|_1 \log m \log n}{\left(\frac{e^{2\gamma}-1}{e^{2\gamma}+1}\right)^2 \max_{ij} a_{ij}^2} + m + \frac{\log n}{\beta} + n \right)$ taking into account the time required to compute $W$ ($O(m)$) and $\tilde{S}_{\Delta,\beta}(G)$ ($O(\log(n)/\beta + n)$) and substituting the values of $c$ and $\theta$ as described in the Algorithm. $\qquad\square$

**Corollary 1.** *Suppose $\max_{ij} a_{ij} > 0$, $G$ is $(C, \Lambda)$-transitive (as in Definition 5), and $\delta \ge n^{-2}$. Then, for any parameter $\gamma > 0$. Algorithm 2 outputs $\tilde{S}_{\Delta,\beta}(G)$, which is a $(\gamma, \delta)$-upper approximation to $S_{\Delta,\beta}(G)$, in $O(m \max(1/C^2, \Lambda^2)) \log m \log n / \min(\gamma^4, 1/4) + n)$ time.*

*Proof.* Since $\left(\frac{e^{2\gamma}-1}{e^{2\gamma}+1}\right)^2 > \gamma^4$ for $0 < \gamma < 1/\sqrt{2}$, we have $\left(\frac{e^{2\gamma}-1}{e^{2\gamma}+1}\right)^2 > \min(\gamma^4, 1/4)$. Next, for any node $i$ and $j \in \mathcal{N}_i$, we have $\frac{deg(j)}{\max_{i'j'} a_{i'j'}} \le \max\{\frac{1}{C}, \Lambda\}$: if $deg(j) > \Lambda$, it must be the case that at least $C$-fraction of the paths $i, j, k$ for $k \in \mathcal{N}_j$ are closed, which means $a_{ij} \ge C \cdot deg(j)$, and so $\frac{deg(j)}{\max_{i'j'} a_{i'j'}} \le \frac{1}{C}$; if $deg(j) \le \Lambda$, we have $\frac{deg(j)}{\max_{i'j'} a_{i'j'}} \le \Lambda$. Therefore, $\frac{\|W\|}{\max_{ij} a_{ij}^2} = \sum_{i \in V(G)} \sum_{j \in \mathcal{N}_i} \frac{deg(i) \times deg(j)}{\max_{i'j'} a_{i'j'}^2} = \sum_{i \in V(G)} \frac{deg(i)}{\max_{i'j'} a_{i'j'}} \sum_{j \in \mathcal{N}_i} \frac{deg(j)}{\max_{i'j'} a_{i'j'}} \le \sum_{i \in V(G)} \sum_{j \in \mathcal{N}_i} \max(\frac{1}{C^2}, \Lambda^2) = 2m \max(\frac{1}{C^2}, \Lambda^2)$, the Corollary follows. $\qquad\square$

## 5 Higher-order local sensitivity and improved bounds for path counting

As mentioned earlier in Section 2, Karwa et al. [23] develop a different technique involving local sensitivity of the local sensitivity, and use it for private counts of $k$-cliques, since it seems hard to

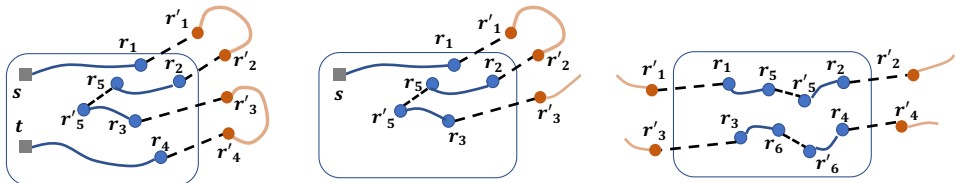

Figure 2: Exaples of compact subgraphs and anchor edges in the algorithm for computing higher-order sensitivity of paths: (Left) $D = \{r_1, r_2, r_3, r_4\}$, edges $(r_1, r_1')$, $(r_2, r_2')$, $(r_3, r_3')$ and $(r_4, r_4')$ are external anchor edges, while $(r_5, r_5')$ is an internal anchor edge. Nodes $s$ and $t$ are the starting and ending nodes of the path; (Middle) $D = \{r_1, r_2, r_3\}$, and $(r_1, r_1')$, $(r_2, r_2')$, $(r_3, r_3')$ are external anchor edges, while $(r_5, r_5')$ is an internal anchor edge. Node $s$ is the starting node of the path, and the path ends at some other node, not in this compact subgraph; (Right) $D = \{r_1, r_2, r_3, r_4\}$, edges $(r_1, r_1')$, $(r_2, r_2')$, $(r_3, r_3')$ and $(r_4, r_4')$ are external anchor edges, while $(r_5, r_5')$ and $(r_6, r_6')$ are internal anchor edges. Paths pass through this compact subgraph, without starting or ending in it.

estimate smooth sensitivity for such subgraphs. We generalize it, and develop a multi-level local sensitivity approach, extending [23], for privately counting the number of copies of any subgraph $H$; as in the case of smooth sensitivity, we show that this also works with approximate queries.

Let $\ell$ denote the number of edges in $H$. Our main idea involves finding private bounds for higher-order local sensitivity. Let $L = \{\{i, j\} : i, j \in V\}$ denote all possible pairs of nodes in $V$. For subgraph $H$, and a subset $S \subset \{\{i, j\} : i, j \in V\}$ of pairs of nodes, let $f_H(G, S)$ denote the number of embeddings of $H$ in the graph $G(V, E \cup S)$, which *contain all the edges corresponding to pairs of nodes in $S$*. For $S = \emptyset$, $f_H(G, \emptyset)$ is the number of embeddings of $H$ in $G$. We drop the subscript $H$ when it is clear from the context. Let $f^{(k)}(G) = \max_{|S|=k} f(G, S)$; note that $f^{(0)}(G) = f(G)$. It can be seen that $LS(f^{(k)}) \leq f^{(k+1)}(G)$, which is the basis for considering higher-order local sensitivity for subgraph counting. However, a careful analysis of the privacy bounds becomes involved. Our approach is described in Algorithm 5 in the supplementary material, and the main result is summarized below.

**Theorem 3.** *Let $g^{(k)}(G) = f^{(k)}(G) + g^{(k+1)}(G)\frac{\ln 1/\delta_2}{\epsilon_2} + Lap(g^{(k+1)}(G)/\epsilon_2)$, for $k = \ell - 1, \cdots, 0$ as computed in Algorithm 5. Then, $g^{(0)}(G)$ is an $((k_m + 1)\epsilon_2, \delta_2 + (k_m + 1)e^{\epsilon_2}\delta_2)$-DP estimate of $f(G)$.*

The above result, and the definition of $f^{(k)}(G)$ implies that for any subgraph with $\ell$ edges, the higher-order local sensitivity approach can be run in time $O(n^{2\ell})$. This can be improved further for paths and trees, as summarized below.

**Lemma 6.** *Private counting for a graph $H$ with a constant $\ell$ number of edges using higher-order local sensitivity (Algorithm 5) has running time $O(n^{2\ell})$. When $H$ is a path or tree with $\ell$ edges, Algorithm 5 has running time $O(n^{\ell+1})$.*

**Higher-order local sensitivity with approximate queries.** We show that our method works even if we have a very good approximation $\hat{f}(x)$ to $f(x)$, instead of the exact value (Theorem 7 in the supplementary material). This allows us to use $\hat{f}(x)$ instead of $f(x)$ in Algorithm 5.

**Improved bounds for path counting in graphs with bounded degeneracy and treewidth.** We say that $G$ has degeneracy $d$, if the nodes can be ordered so that each node has at most $d$ neighbors with higher index [13]. Informally, $G$ has treewidth $t$ if it has recursive balanced separators of size $t$; see [11] for details. There has been a lot of work showing that non-private algorithms for subgraph counting can be done efficiently when $G$ has either of these parameters bounded, e.g., [5, 13, 6, 11]. We show that if $G$ has degeneracy $d$ and treewidth $t$, then the higher-order local sensitivity of paths of length $k$ can be computed in time $O(h(d, t, k)poly(n))$, where $h(d, t, k)$ is independent of $n$; thus the running time does not scale as $n^k$.

Our algorithm and analysis are summarized in Section E in the supplementary material (Theorem 8). Recall that the goal is to compute $f^{(k)}(G) = \max_{S:|S| \leq k} f(G, S)$. We summarize the main ideas here. We show that a feasible solution, which involves fixing a subset $S$ of pairs of nodes, can be viewed as a tuple of at most $k$ disjoint subgraphs $(H_i, D_i)$, referred to as $k$-compact subgraphs. This is

| Network | Description | #nodes | #edges | #triangles |
|---------|-------------|--------|--------|------------|
| Oregon-1 | AS peering information-Oregon route-views | 10,670 | 22,002 | 17,145 |
| ca-HepTh | Collaboration network-Arxiv High Energy Physics | 9,877 | 51,971 | 28,339 |
| ca-GrQc | Collaboration network-Arxiv General Relativity | 5,242 | 14,496 | 48,620 |
| Oregon-2 | AS peering information-Oregon route-views | 10,900 | 31,h180 | 82,857 |
| ca-CondMat | Collaboration network-Arxiv Condensed Matter | 23,133 | 186,936 | 173,361 |
| loc-Brightkite | Brightkite location based online social network | 58,228 | 428,156 | 494,728 |
| com-Amazon | Amazon product network | 334,863 | 925,872 | 667,129 |
| email-Enron | Email communication network- Enron | 36,692 | 367,662 | 727,044 |
| ca-AstroPh | Collaboration network-Arxiv Astro Physics | 18,772 | 198,110 | $1.35 \times 10^6$ |
| com-DBLP | DBLP collaboration network | 317,080 | 2,099,732 | $2.22 \times 10^6$ |
| loc-Gowalla | Gowalla location based online social network | 196,591 | 950,327 | $2.27 \times 10^6$ |
| ca-HepPh | Collaboration network-Arxiv High Energy Physics | 12,008 | 237,010 | $3.36 \times 10^6$ |

Table 2: Statistics of tested networks.

defined to be a subgraph which satisfies the following conditions (Figure 6): (1) $H = G_\sigma(v, k, F^{(i)})$ for some node $v$ and subset $F^{(i)} = \{e_1, \ldots, e_i\}$, where $G_\sigma$ is the $k$-hop graph starting at node $v$, only restricted to higher degeneracy order nodes than $v$, (2) the $k$-hop neighborhood is expanded as edges in $F^{(i)}$ (referred to as internal anchor edges) are added, one at a time, and (3) $D$ is a subset of vertices in $H$, and is referred to as the set of connectors. The compact subgraphs are connected via edges between connector nodes.

We show that paths using edges from $S$ (which contribute to the count $f(G, S)$) can be viewed to consist of segments within these compact subgraphs, passing through internal anchor edges, as well as edges between connectors (which are also from $S$). We guess the number of path segments between connectors within a factor of $(1 + 1/(\log n)^c)$ for a constant $c$, which allows us to search for a tuple of compact subgraphs with the corresponding approximate counts. We show that the number of such tuples is $O(h(d, k)poly(n))$, and existence of given tuple can be determined in a similar time.

## 6   Experimental Results

We evaluate our algorithm for privately computing #triangles using *approximate* smooth sensitivity (Algorithm 2), compared with the *exact* version of smooth sensitivity and other baselines.

**Datasets.** We consider different real-world networks from [26] as inputs. The networks have have sizes range from $10K - 300K$ nodes, with one of the largest ones having over 2 million edges. Statistics of the networks are presented in Table 2. Due to space limit, we only present the results for some selected networks here. The full results are presented in the SM.

**Infrastructure.** All algorithms are implemented in C++ and OpenMP framework for parallelization. We ran our experiments on a system with a 48-core Intel(R) Xeon(R) Gold 6248R CPU @ 3.00GHz and 1.5TB RAM and limit the number of parallel threads in all experiments to 40.

**Baselines.** We implement two baseline methods: the exact smooth sensitivity of triangle count to calibrate Laplacian noise (denoted in our experiment as Karwa et al.) [23], and using the ladder function method (Zhang et al.) [36].

**Metrics for evaluating performance.** For each input network $G$, we calculate the following metrics:
• Accuracy of the private triangle counts, defined as TRIANGLE COUNT RELATIVE ERROR = $\frac{|M_{f_\Delta}(G) - f_\Delta(G)|}{f_\Delta(G)}$, where $M_{f_\Delta}$ is the tested private mechanism to output $f_\Delta(G)$.
• Speedup of Algorithm 2, which compares its running time with the time required for [36, 23] SPEEDUP = $\frac{\text{Runtime of our algorithm on } G}{\text{Runtime of baseline algorithm on } G}$ (Figure 5).
• Approximation error of Algorithm 2 for estimating smooth sensitivity, defined as SENSITIVITY APPROXIMATION ERROR = $\frac{\tilde{S}_\beta(G) - S_\beta(G)}{S_\beta(G)}$ (Figure 4).

**Parameters.** We test our algorithms at different privacy budgets $\epsilon \in \{2^{\{-3,-2,-1,0,1\}}\}$ and a fixed $\delta = 10^{-6}$. We report the average accuracy and runtime of five (5) repeats of the private triangle count via approximate smooth sensitivity experiments due to the randomness used in the algorithm. For the exact calculation, we calculate the smooth sensitivity once and re-sample the noise in each run, since

the exact calculation of the sensitivity is deterministic and does not change for each setting. We use the true counts of triangles of the networks as reported by [26].

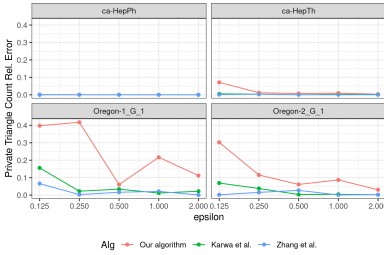 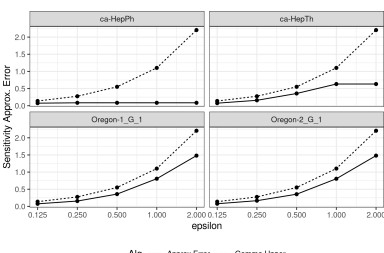

Figure 3: Triangle Count Relative Error, showing the accuracy of the private triangle count with noise calibrated by (1) our approximate smooth sensitivity (Algorithm 2), (2) the exact smooth sensitivity (Karwa et al.), and (3) the ladder function (Zhang et al.)

Figure 4: Error factor in approximate smooth sensitivity output by Algorithm 2 (relative to the exact smooth sensitivity) on selected networks. Dotted lines indicate the theoretical upper bound factor $e^{\gamma}$ (guaranteed by Theorem 2).

**Experimental Results.** The results show that our algorithm can achieve similar levels of accuracy to the exact calculation while being several orders of magintude faster. Figure 3 shows that when using approximate smooth sensitivity, the private triangle counts reach the accuracy of using the exact calculation of the smooth sensitivity across different privacy budgets in all but two networks (Oregon 1 and Oregon 2). In these two networks, the smooth sensitivities are relatively large in comparison with the triangle counts, which makes the noise fluctuate more for the approximate estimate (see Figure 8).

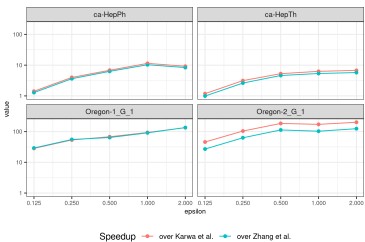

Figure 5: Speedup of Algorithm 2 over the exact smooth sensitivity (Karwa et al.), and the ladder function (Zhang et al.)) on selected networks.

Figure 4 shows that all approximate smooth sensitivities are within some small factor of the exact smooth sensitivities. Noter that approximate smooth sensitivities are always larger than the true smooth sensitivity. It is important since a lower value may expose the privacy due to an inadequate noise magnitude. In general, a smaller value of $\epsilon$ (higher privacy guarantee) requires a more accurate estimation of the sensitivity. It is illustrated in Figure 4 as the approximate smooth sensitivity is close to the exact smooth sensitivity in all networks at $\epsilon = 0.125$ or $\epsilon = 0.25$.

Figure 5 shows that our approximation algorithm is orders of magnitude faster than the exact algorithm. In large graphs (Amazon, DBLP, Gowalla), the speedup may reach $1,000$-fold. Generally, a lower $\epsilon$ requires a smaller approximation factor ($\gamma$), which in turn requires a larger number of iterations to reduce the error in approximation. The majority of tested networks have speedup factors between 10 and 100-fold across all privacy budgets $\epsilon$.

## 7    Conclusions and future work

We give significant improvement in the running time for privately counting the number of embeddings of constant size subgraphs in a graph, without using noise based on global sensitivity. Despite a lot of work on private subgraph counting, efficient algorithms were not known for many subgraphs. Our results for triangles show significant benefits of our approach, by improving on the runtime over all prior private algorithms. Our approach of using approximations to sensitivity metrics opens the possibility of using other techniques from graph sampling and sketching to obtain more efficient algorithms. For general subgraphs, our focus here has been on theoretical bounds using multi-level local sensitivity combined with approximate queries. Developing efficient and practical algorithms for these problems is a good direction for future work.

## Acknowledgments and Disclosure of Funding

This work was partially supported in part by the following grants: National Science Foundation Grants CCF-1918656 (Expeditions), OAC-1916805 (CINES), IIS-1955797, and CNS-2317193, VDH Grant PV-BII VDH COVID-19 Modeling Program VDH-21-501-0135, DTRA subcontract/ARA S-D00189-15-TO-01-UVA, NIH 2R01GM109718-07, CDC MIND cooperative agreement U01CK000589, and the U.S. DOE Exascale Computing Project's (ECP) (17-SC-20-SC) ExaGraph codesign center at Pacific Northwest National Laboratory.

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

## A  Differential Privacy with high probability

Following [24], we define Indistinguishability and Point-wise Indistinguishability as follows:

**Definition 8.** *Two random variables $X, Y$ having the same support are $(\epsilon, \delta)$-indistinguishable if for all set $O \subseteq Supp(X)$,*

$$\Pr[X \in O] \leq e^\epsilon \Pr[Y \in O] + \delta \text{ and } \Pr[Y \in O] \leq e^\epsilon \Pr[Y \in O] + \delta$$

**Definition 9.** *Two random variables $X, Y$ are point-wise $(\epsilon, \delta)$-indistinguishable if with probability at least $1 - \delta$ over a drawn from either $X$ or $Y$, we have:*

$$e^{-\epsilon} \Pr[Y = a] \leq \Pr[X = a] \leq e^\epsilon \Pr[Y = a]$$

The following lemma allows us to interpret a mechanism that is $\epsilon$-differentially private with probability at least $1 - \delta$ as $(\epsilon, \delta)$-differentially private.

**Lemma 7.** *(Lemma 3.3 of [24]) If $X, Y$ are point-wise $(\epsilon, \delta)$-indistinguishable then they are $(\epsilon, \delta)$-indistinguishable.*

We extend the definition of $(\epsilon, \delta)$-indistinguishability by allowing some small error probability $\delta'$:

**Definition 10.** *Two random variables $X, Y$ having the same Support are $(\epsilon, \delta, \delta')$-indistinguishable if for all set $O \subseteq Supp(X)$, with probability at least $1 - \delta'$ over a drawn from either $X$ or $Y$, we have:*

$$\Pr[X \in O] \leq e^\epsilon \Pr[Y \in O] + \delta \text{ and } \Pr[Y \in O] \leq e^\epsilon \Pr[Y \in O] + \delta$$

The following lemma allows us to interpret a mechanism that is $(\epsilon, \delta)$-differentially private with probability at least $1 - \delta'$ as $(\epsilon, \delta + \delta')$-differentially private.

**Lemma 8.** *If $X, Y$ are $(\epsilon, \delta, \delta')$-indistinguishability then they are $(\epsilon, \delta + \delta')$-indistinguishable.*

*Proof.* Our proof is insired by [24].

We define the set $Bad$ as follow:

$$Bad = \{O : \Pr[X \in O] > e^\epsilon \Pr[Y \in O] + \delta \text{ or } \Pr[Y \in O] > e^\epsilon \Pr[Y \in O] + \delta\}$$

Since $X, Y$ are $(\epsilon, \delta, \delta')$-indistinguishability, $\Pr[X \in Bad] < \delta'$. We have:

$$\begin{aligned}
\Pr[X \in O] &\leq \Pr[X \in O \setminus Bad] + \Pr[X \in Bad] \\
&\leq e^\epsilon \Pr[Y \in O \setminus Bad] + \delta + \delta' \\
&\leq e^\epsilon \Pr[Y \in O] + \delta + \delta'.
\end{aligned}$$

Similarly for $Y$, we have $\Pr[Y \in O] \leq e^\epsilon \Pr[X \in O] + \delta + \delta'$ and the Lemma follows.  $\square$

## B  Missing proofs of Section 4

**Approximate Smooth Sensitivity for exact queries**

When we can calculate a $\gamma$-approximation of the smooth sensitivity $S^*_{f,\beta}$, i.e., $\tilde{S}_{f,\beta}$ is a bounded approximation of $S^*_{f,\beta}$ with probability 1, we can substitute $S^*_{f,\beta}$ by $\tilde{S}_{f,\beta}$ to calibrate the noise directly, although with a bit larger effective constant $\beta' = \beta + \gamma$, which in turn will make the noise a bit larger. Lemma 9 formalizes the equivalence of $S^*_{f,\beta}$ and $\tilde{S}_{f,\beta}$ in this case.

**Lemma 9.** *If $\tilde{S}_{f,\beta}$ is a $\gamma$-approximation of the smooth sensitivity $S^*_{f,\beta}$, then $\tilde{S}_{f,\beta}$ is a $\beta'$-smooth sensitivity of $f$ at $\beta' = \beta + \gamma$.*

*Proof.* By definition of $\tilde{S}_{f,\beta}$ and $S^*_{f,\beta}$, it follows that $\tilde{S}_{f,\beta}(D) \geq S^*_{f,\beta}(D) \geq LS_f(D)$ for all $x$.

Second, for all $D \sim D' \in D^n$, we have: $\tilde{S}_{f,\beta}(D) \leq e^\gamma S^*_{f,\beta}(D) \leq e^\gamma e^\beta S^*_{f,\beta}(D') \leq e^{\gamma+\beta}\tilde{S}_{f,\beta}(D') \leq e^{\beta'}\tilde{S}_{f,\beta}(D')$ , the lemma follows from Definition 2. $\qquad\square$

In the case when we can only calculate a $(\gamma, \delta')$-approximation of the smooth sensitivity, i.e., $\tilde{S}_{f,\beta}$ is a $\gamma$-approximation of $S^*_{f,\beta}$ with probability at least $1 - \delta'$, Lemma 9 cannot be directly applied. Similar to [31], we use the formalism of admissible noise, to prove we can use $\tilde{S}_{f,\beta}$ to calibrate an $(\alpha, \beta)$-admissible noise with appropriate values of $\alpha$ and $\beta$ to deliver the privacy guarantee. In Lemma 10, we analyze the output of such a mechanism on an arbitrary pair of neighbors $D$ and $D'$. We condition our analysis on the events of success of $\tilde{S}_{f,\beta}$ on both $D$ and $D'$ (with probability at least $1 - 2\delta'$). The failure probability (at most $2\delta'$) will then add up to the $\delta$-part of the final $(\epsilon, \delta)$-DP guarantee.

**Lemma 10.** *Let $h$ be $(\alpha, \beta')$-admissible noise. Let $\tilde{S}_{f,\beta}$ be a $(\gamma, \delta')$-approximation to $S^*_{f,\beta}$, where $\beta = \beta' - \gamma$. Then, $A(D) = f(D) + \frac{\tilde{S}_{f,\beta}(D)}{\alpha}Z$ is $(\epsilon, \frac{e^{\epsilon/2}+1}{2}\delta + 2\delta')$-differentially private, where $Z \sim h$.*

*Proof.* We prove that with probability at least $1 - 2\delta'$, $A(D)$ is $(\epsilon, \frac{e^{\epsilon/2}+1}{2}\delta)$-differentially private, and by applying Lemma 8 the Lemma follows.

Let $\mathcal{E}$ be the event that $S^*(D) \leq \tilde{S}_{f,\beta}(D) \leq e^\gamma S^*(D)$; we have $\Pr[\mathcal{E}] \geq 1 - \delta'$. Similarly, let $\mathcal{E}'$ be the event that $S^*(D') \leq \tilde{S}_{f,\beta}(D') \leq e^\gamma S^*(D')$ for a fixed neighbor $D'$ of $D$; we have $\Pr[\mathcal{E}'] \geq 1 - \delta'$, and $\Pr[\mathcal{E} \cup \mathcal{E}'] \geq 1 - 2\delta'$. We will condition on $\mathcal{E} \cup \mathcal{E}'$.

Let $\mathcal{S} \subset \mathbb{R}^+$. Let $N(D) = \frac{\tilde{S}_{f,\beta}(D)}{\alpha}$, and define $\mathcal{S}_1 = \frac{\mathcal{S}-f(D)}{N(D)}$, $\mathcal{S}_2 = \mathcal{S}_1 + \frac{f(D)-f(D')}{N(D)} = \frac{\mathcal{S}-f(D')}{N(D)}$, $\mathcal{S}_3 = \mathcal{S}_2 \frac{N(D)}{N(D')} = \frac{\mathcal{S}-f(D')}{N(D')}$.

We observe that $A(D) \in \mathcal{S}$ if and only if $Z \in \mathcal{S}_1$. Similarly, $A(D') \in \mathcal{S}$ if and only if $Z \in \mathcal{S}_3$. We have

$$\Pr[A(D) \in \mathcal{S}] = \Pr[Z \in \mathcal{S}_1] \leq e^{\epsilon/2} \Pr\left[Z \in \mathcal{S}_1 + \frac{f(D)-f(D')}{N(D)}\right] + \delta/2$$

$$= e^{\epsilon/2}\Pr[Z \in \mathcal{S}_2] + \delta/2 \leq e^\epsilon \Pr\left[Z \in \mathcal{S}_2 \exp\left(\ln\frac{N(D)}{N(D')}\right)\right] + e^{\epsilon/2}\delta/2 + \delta/2$$

$$= e^\epsilon \Pr[Z \in \mathcal{S}_3] + e^{\epsilon/2}\delta/2 + \delta/2 = e^\epsilon \Pr[A(D') \in \mathcal{S}] + (e^{\epsilon/2}+1)\delta/2,$$

where the first inequality is because of Sliding Property and

$$\frac{|f(D)-f(D')|}{N(D)} = \alpha\frac{|f(D)-f(D')|}{\tilde{S}_{f,\beta}(D)} \leq \alpha\frac{|f(D)-f(D')|}{S^*(D)} \leq \alpha,$$

which satisfies the condition for the Sliding Property,

and the second inequality is because of Dilation Property and

$$\left|\ln\frac{N(D)}{N(D')}\right| = \left|\ln\frac{\tilde{S}_{f,\beta}(D)}{\tilde{S}_{f,\beta}(D')}\right| \leq \left|\ln\left(e^\gamma\frac{S^*(D)}{S^*(D')}\right)\right| \leq |\gamma + \ln e^{\pm\beta}| \leq \beta',$$

which satisfies the condition for the Dilation Property.

$\qquad\square$

Lemma 11 specializes Lemma 10 by using Laplacian noise. We note that this is a generalization of [31]'s analysis of Laplacian noise. We introduce parameter $k$ to control the magnitude of the Laplacian noise, while [31]'s analysis fixed it to 1. This design gives us the flexibility to choose the values of $\alpha$ and $\beta$ given fixed targets of $\epsilon$ and $\delta$. When $k = 1/2$, our design yields the same results as [31].

**Lemma 11.** *Let $\alpha = k\epsilon$ and $\beta = \frac{2k^2\epsilon}{\log(2/\delta)}$ for any constant $k > 0$, then $Lap(2k)$ is an $(\alpha, \beta)$-admissible noise.*

*Proof.* For som real-valued random variable $Y$, let $\rho_\delta(Y)$ be the least solution to $\Pr[Y \le \rho_\delta] > 1-\delta$. Let $\delta' = \delta/2$. Let $Z \sim Lap(2k)$. It follows that $|Z| \sim Exponential(1/(2k))$. We will solve the value of $\rho_{\delta'}(|Z|)$ as: $1 - e^{-2k|\rho_{\delta'}(|Z|)|} = 1 - \delta'$ or $\rho_{\delta'}(|Z|) = \frac{\log(2/\delta)}{2k}$.

Substituting $\rho_{\delta'}(|Z|)$ to the formula of $\beta$, we have:

$$\beta = \frac{2k^2\epsilon}{\log(2/\delta)}$$
$$= \frac{k\epsilon}{\rho_{\delta'}(|Z|)}$$

We first prove the Sliding property of $Lap(2k)$. It is equivalent to prove that $\ln(\frac{h(z+\Delta)}{h(z)}) \le \epsilon/2$ for any $|\Delta| < \alpha$ and $h(z)$ is the PDF of $Lap(2k)$. For the distribution $Lap(2k)$, $h(z) = \frac{1}{4k}e^{-|z|/(2k)}$. We have:

$$\ln\left(\frac{h(z+\Delta)}{h(z)}\right) = \ln\left(\frac{\exp(-|z+\Delta|/(2k))}{\exp(-|z|/(2k))}\right)$$
$$= \frac{1}{2k}(|z| - |z+\Delta|)$$
$$\le \frac{1}{2k}|\Delta|$$
$$\le \frac{1}{2k}k\epsilon$$
$$= \frac{\epsilon}{2}.$$

We next prove the Dilation property of $Lap(2k)$, or prove that $\ln(\frac{e^\lambda h(e^\lambda z)}{h(z)}) \le \epsilon/2$ with probability at least $1 - \delta/2$. In the first case, setting $\lambda > 0$, we have $h(e^\lambda z) < h(z)$, and that $\ln(\frac{e^\lambda h(e^\lambda z)}{h(z)}) \le \lambda \le \frac{k\epsilon}{\rho_{\delta'}(|Z|)}$. Since $|Z| \sim Exponential(1/(2k))$, the median of the distribution of $|Z|$ is $\frac{\ln(2)}{1/(2k)} = 2\ln(2)k$. With $\delta < 1$, we have $\delta' < 1/2$ and therefore $\rho_{\delta'}(|Z|) \ge \text{median}(|Z|) = 2ln(2)k$. It follows that $\ln(\frac{e^\lambda h(e^\lambda z)}{h(z)}) \le \frac{k\epsilon}{2\ln(2)k} < \frac{\epsilon}{2}$.

In the second case, $\lambda < 0$. We consider the ratio $\frac{h(e^\lambda z)}{h(z)}$:

$$\frac{h(e^\lambda z)}{h(z)} = \frac{\exp(-|ze^\lambda|/(2k))}{\exp(-|z|/(2k))}$$
$$= \exp\left(\frac{|z|}{2k}(1 - e^\lambda)\right)$$
$$\le \exp\left(\frac{|z|}{2k}|\lambda|\right).$$

We then have $\ln(\frac{e^\lambda h(e^\lambda z)}{h(z)}) \le \lambda + \frac{|z||\lambda|}{2k} < \frac{|z||\lambda|}{2k}$. Consider the event $G : \{z : |z| < \rho_{\delta'}(|Z|)\}$. Under $G$, we have $|z||\lambda|\frac{1}{2k} \le \rho_{\delta'}(|Z|)\frac{\epsilon}{\rho_{\delta'}(|Z|)}\frac{1}{2k} = \frac{\epsilon}{2}$. Under $h$, $\Pr[G] = 1 - \delta/2$, which completes the proof for the Dilation property of $Lap(2k)$.

$\square$

Finally, we apply a Laplacian noise with appropriate parameters to provide a concrete mechanism using Approximate Smooth Sensitivity $\tilde{S}_{f,\beta}$ to guarantee $(\epsilon, \delta)$-DP.

**Lemma 12.** *(Lemma 2) Let $\tilde{S}_{f,\beta}$ be a $(\gamma, \delta')$-approximation to $S^*_{f,\beta}$, for $\gamma = \beta = \frac{16\epsilon}{\ln(2/\delta)}$. Then,*
$A(D) = f(D) + Lap(\frac{2\tilde{S}_{f,\beta}(D)}{\epsilon})$ *is $(\epsilon, \frac{e^{\epsilon/2}+1}{2}\delta + 2\delta')$-differentially private.*

*Proof.* Set $k = 4$, using Lemma 11 with $\alpha = k\epsilon = 4\epsilon$, and $\beta' = \beta + \gamma = \frac{32\epsilon}{\ln(2/\delta)} = \frac{2k^2\epsilon}{\ln(2/\delta)}$. Then, from Lemma 11, the $Lap(2k) = Lap(8)$ distribution is $(\alpha, \beta')$-admissible. By Lemma 10, $A(D) = f(D) + \frac{\tilde{S}_{f,\beta}(D)}{\alpha}Z = f(D) + \frac{\tilde{S}_{f,\beta}(D)}{4\epsilon}Lap(8) = f(D) + Lap(2\tilde{S}_{f,\beta}(D)/\epsilon)$ is $(\epsilon, \frac{e^{\epsilon/2}+1}{2}\delta + 2\delta')$-differentially private and the corollary follows. $\square$

**Approximate Smooth Sensitivity for Approximate queries**

**Definition 11.** *(Definition 7) We define a functions $g_f$ and $s_f$ as:*

$$g_f = \begin{cases} A_f(D) \text{ if } (1-\alpha_1)f(D) \le A_f(D) \le (1+\alpha_1)f(D), \\ (1-\alpha_1)f(D) \text{ if } A_f(D) < (1-\alpha_1)f(D), \\ (1+\alpha_1)f(D) \text{ if } A_f(D) > (1+\alpha_1)f(D). \end{cases} \tag{3}$$

$$s_f = \begin{cases} \tilde{S}_{f,\beta}(D) \text{ if } S_{f,\beta}(D) \le \tilde{S}_{f,\beta}(D) \le e^\gamma S_{f,\beta}(D), \\ S_{f,\beta}(D) \text{ if } \tilde{S}_{f,\beta}(D) < S_{f,\beta}(D), \\ e^\gamma S_{f,\beta}(D) \text{ if } \tilde{S}_{f,\beta}(D) > e^\gamma S_{f,\beta}(D). \end{cases} \tag{4}$$

*Let $S_{g_f}(D) = 4\alpha_1 g_f(D) + 2s_f(D)$*

---

**Algorithm 3** Fast algorithm using Approximate Smooth Sensitivity for Approximate Query $A_f$

**Input:** $G, f, \epsilon, \delta$
**Output:** $(\epsilon, (e^{\epsilon/2} + 2)\delta)$-differentially private estimaton of query $f$

1: $\beta := \gamma := 4\alpha_1 = \frac{\epsilon}{6\log(2/\delta)}$
2: $\delta_1 := \delta_2 = \delta/2$
3: Calculate the $(\gamma, \delta_2)$-upper approximate smooth sensitivity $\tilde{S}_{f,\beta}(G)$
4: Calculate the $(\alpha_1, \delta_1)$-approximate $f$ $A_f(G)$
5: **Return** $A_f(G) + \frac{8\alpha_1 A_f(G) + 4\tilde{S}_{f,\beta}(G)}{\epsilon}Lap(1)$

---

Lemma 13 proves that $S_{g_f}$ is a $\beta'$-smooth upper bound of the local sensitivity of $g_f$, i.e., we have to prove the two conditions:

- $LS_{g_f}(D) \le Sg_f(D)$, and

- $S_{g_f}(D) \le e^{\beta'} S_{g_f}(D')$ for any pair of neighbor datasets $D \sim D'$,.

where $LS_{g_f}(D) = max_{D' \sim D}|g_f(D) - g_f(D')|$ or the local sensitivity of $g_f$ at $D$.

**Lemma 13.** *(Lemma 3) Given $g_f$, $s_f$, and $S_{g_f}$ as defined above, $S_{g_f}$ is a $\beta'$-smooth upper bound of the local sensitivity of $g_f$, where $\beta' \ge 4\alpha_1 + \gamma + \beta$ and $\alpha_1 < 1/2$.*

*Proof.* We now prove the two conditions. We first start with proving $S_{g_f}$ is an upper bound on the local sensitivity $LS_{g_f}$ of $g_f$:

$$
\begin{aligned}
LS_{g_f}(D) &= \max_{D':D'\sim D} \|g_f(D) - g_f(D')\| \\
&\leq \max_{D':D'\sim D} \|(1+\alpha_1)f(D) - (1-\alpha_1)f(D')\| \\
&\leq \max_{D':D'\sim D} \alpha_1\|f(D) + f(D')\| + \|f(D) - f(D')\| \\
&\leq \alpha_1\|f(D) + f(D) + LS(D)\| + LS_f(D) \\
&\leq 2\alpha_1 f(D) + 2LS_f(D) \\
&\leq 4\alpha_1 g_f(D) + 2S_{f,\beta}(D) \\
&\leq 4\alpha_1 g_f(D) + 2s_f(D) \\
&= S_{g_f}(D).
\end{aligned}
$$

The first inequality uses Definition 7 and the fifth uses the facts, $f(D) \leq \frac{A_f(D)}{1-\alpha_1} \leq 2g_f(D)$, and $LS_f(D) \leq S_{f,\beta}(D)$ (as $S_{f,\beta}(D)$ is a upper bound on $LS_f(D)$).

For the second condition, we need to prove that for any pair $D \sim D'$, the values of $S_{g_f}$ on them do not differ by more than a multiplicative factor of $e^{\beta'}$ with $\beta' = 4\alpha_1 + \gamma + \beta$.

$$
\begin{aligned}
S_{g_f}(D) &= 4\alpha_1 g_f(D) + 2s_f(D) \\
&\leq 4\alpha_1(1+\alpha_1)f(D) + 2s_f(D) \\
&\leq 4\alpha_1(1+\alpha_1)(f(D') + s_f(D)) + 2s_f(D) \\
&= 4\alpha_1(1+\alpha_1)f(D') + 2(2\alpha_1(1+\alpha_1) + 1)s_f(D) \\
&\leq \frac{4\alpha_1(1+\alpha_1)}{1-\alpha_1}g_f(D') + 2(2\alpha_1(1+\alpha_1) + 1)e^{\gamma+\beta}s_f(D') \\
&\leq 4\alpha_1(1+\alpha_1)(1+2\alpha_1)g_f(D') + 2(1+4\alpha_1)e^{\gamma+\beta}s_f(D') \\
&\leq 4\alpha_1(1+4\alpha_1)g_f(D') + 2(1+4\alpha_1)e^{\gamma+\beta}s_f(D') \\
&\leq e^{4\alpha_1+\gamma+\beta}(4\alpha_1 g_f(D') + 2s_f(D')) \\
&= e^{\beta'}S_{g_f}(D').
\end{aligned}
$$

$\square$

We note that $s_f$, $g_f$, and $S_{g_f}$ are all hypothetical functions, which we may not calculate directly. We need to connect the statement above ($S_{g_f}$ is a Smooth Sensitivity of $g_f$) to the functions we can compute ($A_f$ and $\tilde{S}_{f,\beta}$). Again, we define a hypothetical mechanism $A'$ that takes $D$ as its input. By the definition of $A'$ and the properties of admissible noise, $A'$ is $(\epsilon,\delta)$-DP. Instead of $A'$, we can only calculate $A$, by replacing $g_f$ and $s_f$ by $A_f$ and $\tilde{S}_{f,\beta}$ respectively. Next, we argue that $A$ and $A'$ are, in fact, not too different. Lemma 14 conditions on the probability that $A$ and $A'$ are the same and prove that the privacy guarantee of $A'$ is the same with of $A$ in such condition. The failure probabilities (of $A_f$ and $\tilde{S}_{f,\beta}$ in approximation of $f$ and $S_f$) will add up to the $\delta$-part of the final $(\epsilon,\delta)$-DP.

**Lemma 14.** *Let $h$ be $(\alpha,\beta')$-admissible noise with $\beta' = 4\alpha_1 + \gamma + \beta$. With $A_f$ and $\tilde{S}_{f,\beta}$ defined as above, Algorithm $A(D) = A_f(D) + \frac{4\alpha_1 A_f(D) + 2\tilde{S}_{f,\beta}(D)}{\alpha}Z$ is $(\epsilon, \frac{e^{\epsilon/2}+1}{2}\delta + 2\delta_1 + 2\delta_2)$-differentially private, where $Z \sim h$.*

*Proof.* We prove that with probability at least $1 - 2\delta_1 - 2\delta_2$, $A(x)$ is $(\epsilon, \frac{e^{\epsilon/2}+1}{2}\delta)$-differentially private, and by applying Lemma 8, the Lemma follows.

Let $A'(D) = g_f(D) + \frac{S_{g_f}}{\alpha}Z = g_f(D) + \frac{4\alpha_1 s_f(D) + 2s_f(D)}{\alpha}Z$, as we replace $A_f$ and $\tilde{S}_{f,\beta}$ by $g_f$ and $s_f$ respectively. We note that the use of $g_f$ and $s_f$ is only for the purpose of privacy bound analysis.

With $\beta' > 4\alpha_1 + \gamma + \beta$, Lemma 3 shows that $S_{g_f}$ is a $\beta'$-smooth upper bound on the sensitivity of $g_f$. By Lemma 10, $A'(D) = g_f(D) + \frac{S_{g_f}}{\alpha}Z$ is $(\epsilon, \frac{e^{\epsilon/2}+1}{2}\delta)$-differentially private.

Since with probability at least $1 - \delta_1$, $A_f(D)$ is $g_f(D)$, and at least $1 - \delta_2$, $\tilde{S}_{f,\beta}(D)$ is $s_f(D)$, with probability at least $1 - \delta_1 - \delta_2$, $A(D)$ has the same components as $A'(D)$, hence they are identical. For a fixed neighbor $D' \sim D$, we also have that $A'(D)$ is identical with $A'(D')$ with probability at least $1 - \delta_1 - \delta_2$. By Lemma 8, $A(D), A(D')$ are $(\epsilon, \frac{e^{\epsilon/2}+1}{2}\delta, 2\delta_1 + 2\delta_2)$-indistinguishable. It follows that $A(D)$ is $(\epsilon, \frac{e^{\epsilon/2}+1}{2}\delta + 2\delta_1 + 2\delta_2)$-differentially private.

$\square$

**Theorem 4.** *(Theorem 1) Let $A_f$ be a $(\alpha_1, \delta_1)$ approximation to $f$, $\tilde{S}_{f,\beta}$ be a $(\gamma, \delta_2)$-upper approximation to $S^*_{f,\beta}$, for $\gamma = \beta = \frac{8\epsilon}{\ln(2/\delta)}$, $\alpha_1 = \frac{4\epsilon}{\ln(2/\delta)}$. Then, $A(D) = A_f(D) + Lap(\frac{8\alpha_1 A_f(D) + 4\tilde{S}_{f,\beta}(D)}{\epsilon})$ is $(\epsilon, \frac{e^{\epsilon/2}+1}{2}\delta + 2\delta_1 + 2\delta_2)$-differentially private.*

*Proof.* Set $k = 4$, using Lemma 11 with $\alpha = k\epsilon = 4\epsilon$, and $\beta' = \beta + \gamma + 4\alpha_1 = \frac{32\epsilon}{\ln(2/\delta)} = \frac{2k^2\epsilon}{\ln(2/\delta)}$. Then, from Lemma 11, the $Lap(2k) = Lap(8)$ distribution is $(\alpha, \beta')$-admissible. By Lemma 14, $A(D) = f(D) + \frac{\tilde{S}_{f,\beta}(D)}{\alpha}Z = f(D) + \frac{\tilde{S}_{f,\beta}(D)}{4\epsilon}Lap(8) = f(D) + Lap(\frac{2\tilde{S}_{f,\beta}(D)}{\epsilon})$ is $(\epsilon, \frac{e^{\epsilon/2}+1}{2}\delta + 2\delta_1 + 2\delta_2)$-differentially private and the Theorem follows. $\square$

## C  Missing proofs of Section 4.1

The main target of this section is to calculate an approximation of $\max a_{ij}^2$, such that we can apply it to approximate a $\gamma$-approximation of $S_{\Delta,\beta}$ for any $\gamma$ and $\beta$. There are three key components of the analysis:

- Algorithm 4 uses diamond sampling to quickly calculate $x_{ij}$ that approximates $a_{ij}^2$ for every $i, j$. We use this algorithm as an important sub-routine in Algorithm 1 and 2. Lemma 15 shows the utility of the algorithm. We note that due to the Lemma, the quality of each estimation $x_{ij}$ depends on the true value $a_{ij}^2$ itself–which is unknown and is the value we are trying to get. Hence, we cannot directly apply Algorithm 4 to get any-approximation of $\max a_{ij}^2$ as we need.

- Algorithm 1 uses multiple rounds of Algorithm 4 with increasing numbers of iterations. The main idea behind this process is that, with a small number of iterations, Algorithm 4 executes very quickly and gives us a rough estimation $x_{ij}$ of $a_{ij}^2$ for every $i, j$. We then check if the estimation is good enough. We know the estimation is considered good when $\max_{ij} x_{ij} < \frac{3}{2}\tau_k \times$ *{some constants}*. Corollary 2, 3 will explain why. Finally, Lemma 18 concludes the utility of the output of the algorithm, such that after the last iteration $k$, $\tau_k$ is at most 4 times the value of $\max a_{ij}^2$.

- Algorithm 2 further improves the output of Algorithm 1. Now we know roughly how large $\max_{ij}^2$ is, but only within a factor of 4. We will run the final instance of Algorithm 4 with a special constant $\theta$, which we use to control the approximation of the final estimation. $\theta$ is set by $\gamma$-the approximation level required for $\gamma$-approximation of the smooth sensitivity described by the previous section (which in turn is set by $(\epsilon, \delta)$). Lemma 19 concludes the final utility of our estimation.

We define the norm of matrix $A$ as follows: $\|A\| = \|A\|_1 = \sum_{ij} A_{ij}$, $\|A_i\| = \sum_k A_{ik}$. Note that degree $d(i) = \|A_i\|$, $\bar{d}(i) = \|1 - A_i\|$. The following concentration bound is shown in [2] for the output of Algorithm 4.

**Lemma 15.** *Lemma 3 of [2]. Fix $\theta > 0$ and error probability $\beta$. For any $a_{ij} > 0$, if the number of samples $s > 3\|W\| \log(2/\beta)/(\theta^2 a_{ij}^2)$ then*

$$\Pr\left[\left|\frac{x_{ij}\|W\|}{s} - a_{ij}^2\right| > \theta a_{ij}^2\right] \le \beta. \tag{5}$$

The following two lemmas and corollaries relate upper and lower bounds on the random variable $\max_{ij} x_{ij}$ output by Algorithm 4 to the quantity of interest, $\max_{ij} a_{ij}^2$.

---

**Algorithm 4** Fast estimation of $(a_{ij})^2$

**Input:** $G(V, E)$, Number of iterations $s$

**Output:** $x_{ij}$ that approximates $a_{ij}^2$, for all $i, j$

---

1: Let $A$ be the adjacency matrix of $G$
2: **for** $(k, i) \in E$ **do**
3:     $W_{ki} = d(k)d(i)$
4: **end for**
5: $x \leftarrow 0$
6: **for** $l = 1, \ldots, s$ **do**
7:     Sample $(k, i)$ with probability $W_{ki}/\|W\|_1$
8:     Sample $j$ from $\mathcal{N}_k$
9:     Sample $o$ from $\mathcal{N}_i$
10:     $x_{ij} + = A_{oj}$
11: **end for**
12: Return $x$

---

Lemma 16 states that if we choose a constant $\tau$ that is close but larger than $\max a_{ij}^2$, then after running Algorithm 4 with $s$ iterations, the maximum of $x_{ij}$ is not larger than $\tau \times$ *{some constants}*. Similarly, Lemma 17 states that if we choose a constant $\tau$ that is close but smaller than $\max a_{ij}^2$, then after running Algorithm 4 with $s$ iterations, the maximum of $x_{ij}$ is not smaller than $\tau \times$ *{some constants}*.

With appropriate selections of the constants, Corollary 2 and 2 summarize the two lemmas and describe the condition in which we may find a good value for $\tau$. They allow us to observe the output of an instance of Algorithm 4 and compare the outcome with a pre-determined value of $\tau$. Depending on the sign of the comparision $\max x_{ij} \frac{\|W\|}{s} \gtrless \tau$, we can decide that if we are over-estimating $\tau$ (by Corollary 3) or under-estimating $\tau$ (by Corollary 2).

The two lemmas and their corollaries below form the basis for Algorithm 1 that finds an appropriate value of $\tau$ using binary search.

**Lemma 16.** *For any $\tau$ such that $\max_{ij} a_{ij}^2 < \lambda\tau$ for some constant $0 < \lambda$, there exists a constant $\lambda' > \lambda$ such that with probability at least $1 - \frac{1}{poly(n)}$, $\max_{ij} \frac{x_{ij}\|W\|}{s} \leq \lambda'\tau$.*

*Proof.* Consider any $i, j$. By Chernoff's bound,

$$\Pr[x_{ij} > (1 + \theta')\mathbb{E}[x_{ij}]] \leq \exp\left(-\frac{\theta'^2\mathbb{E}[x_{ij}]}{3}\right).$$

As $\mathbb{E}[x_{ij}] = \frac{a_{ij}^2 s}{\|W\|}$, we have that

$$\Pr[x_{ij}\frac{\|W\|}{s} > (1 + \theta')a_{ij}^2] \leq \exp\left(-\frac{\theta'^2 s a_{ij}^2}{3\|W\|}\right).$$

Setting $\lambda' = \frac{(1+\theta')a_{ij}^2}{\tau}$, we have:

$$1 + \theta' = \frac{\lambda'\tau}{a_{ij}^2}.$$

Therefore, we can lower bound $\theta'$ as follows:

$$\theta' = \frac{\lambda'\tau}{a_{ij}^2} - 1$$

$$= \frac{\lambda'\tau - a_{ij}^2}{a_{ij}^2}$$

$$\geq \frac{\lambda'\tau - \lambda\tau}{a_{ij}^2}$$

$$= \frac{\tau(\lambda' - \lambda)}{a_{ij}^2} \text{ since } a_{ij}^2 > \lambda\tau.$$

Now, we substitute $(1 + \theta')a_{ij}^2$ by $\lambda'\tau$:

$$\Pr[x_{ij}\frac{\|W\|}{s} > \lambda'\tau] \leq \exp\left(-\frac{\theta'^2 s a_{ij}^2}{3\|W\|}\right)$$

$$= \exp\left(-\frac{\theta'^2 3\|W\|c\log n a_{ij}^2}{3\|W\|\theta^2\tau}\right)$$

$$= \exp\left(-\frac{\theta'^2 a_{ij}^2 c\log n}{\theta^2\tau}\right)$$

$$\leq \exp\left(-\frac{\tau^2(\lambda'-\lambda)^2 a_{ij}^2 c\log n}{a_{ij}^4\theta^2\tau}\right)$$

$$= \exp\left(-\frac{\tau(\lambda'-\lambda)^2 c\log n}{a_{ij}^2\theta^2}\right)$$

$$\leq \exp\left(-\frac{(\lambda'-\lambda)}{\lambda\theta^2}c\log n\right)$$

$$= n^{-c\frac{(\lambda'-\lambda)^2}{\lambda\theta^2}}.$$

Selecting $\lambda' > \lambda$ s.t $c\frac{(\lambda'-\lambda)^2}{\lambda\theta^2} \geq 3$ and taking the union bound on all pairs $ij$, the Lemma follows. $\qquad\square$

**Corollary 2.** *With $c > 3, \theta = \frac{1}{2}, \lambda = 1, \lambda' = \frac{3}{2}$, with probability at least $1 - 1/n^{c-2}$, if $\max_{ij} x_{ij}\frac{\|W\|}{s} > \frac{3}{2}\tau$ then $\max_{ij} a_{ij}^2 > \tau$.*

**Lemma 17.** *For any $\tau$ such that $\max_{ij} a_{ij}^2 > \lambda\tau$ for some constant $0 < \lambda$, then there exists a constant $\lambda'' \leq (1-\theta)\lambda$ such that with probability at least $1 - \frac{1}{poly(n)}$, $\max_{ij}\frac{x_{ij}\|W\|}{s} \geq \lambda''\tau$.*

*Proof.* Since $\max_{ij} a_{ij}^2 > \lambda\tau$, there exists pairs of nodes $i'j'$ such that $a_{i'j'}^2 > \lambda\tau$. For such pairs $i'j'$, we have:

$$\Pr\left[x_{i'j'}^2\frac{\|W\|}{s} < (1-\theta)a_{i'j'}^2\right] \leq \exp\left(-\frac{\theta^2 s a_{i'j'}^2}{3\|W\|}\right)$$

$$= \exp\left(-\frac{3\theta^2\|W\|a_{i'j'}^2 c\log n}{3\|W\|\theta^2\tau}\right)$$

$$= \exp\left(-\frac{a_{i'j'}^2 c\log n}{\tau}\right)$$

$$\leq \exp\left(-\lambda c\log n\right) = n^{-c\lambda}.$$

Setting $\lambda'' \leq (1-\theta)\lambda$, we have $\lambda'' \leq \frac{(1-\theta)a_{i'j'}^2}{\tau}$, or $\lambda''\tau \leq (1-\theta)a_{i'j'}^2$. Therefore $\Pr\left[x_{i'j'}^2\frac{\|W\|}{s} < \lambda''\tau\right] \leq \Pr\left[x_{i'j'}^2\frac{\|W\|}{s} < (1-\theta)a_{i'j'}^2\right] \leq n^{-c\lambda}$. Choosing $c$ such that $c\lambda > 3$ and taking the union bound over all pairs $i'j'$, the Lemma follows. $\qquad\square$

**Corollary 3.** *With $c > 3, \theta = \frac{1}{2}, \lambda = 3, \lambda'' = \frac{3}{2}$, with probability at least $1 - 1/n^{c-2}$, if $\max_{ij} x_{ij}\frac{\|W\|}{s} < \frac{3}{2}\tau$ then $\max_{ij} a_{ij}^2 < 3\tau$.*

**Lemma 18.** *(Lemma 4) Let $\tau_k$ is the output of Algorithm 1, with probability at least $1 - 1/n^{c-3}$, $\tau_k < \max_{ij} a_{ij}^2 < 4\tau_k$.*

*Proof.* Assume the Algorithm 1 stops at step $k$. At step $k-1$, $\max_{ij} x_{ij} < \frac{3}{2}\tau_{k-1}$, then by Corollary 3 we have $\max_{ij} a_{ij}^2 < 3\tau_{k-1} = 3\frac{4}{3}\tau_k = 4\tau_k$. At step $k$, $\max_{ij} x_{ij} > \frac{3}{2}\tau_k$, then by

Corollary 2 we have $\max_{ij} a_{ij}^2 > \tau_k$. Applying union bound on $k$ steps, with probability at least $1 - 1/n^{c-3}$ we have $\tau_k < \max_{ij} a_{ij}^2 < 4\tau_k$, the Lemma follows. $\qquad\square$

Finally, in Algorithm 2, we postprocess the output of Algorithm 1 to obtain an accurate estimate of $\max_{ij} a_{ij}^2$ and hence, the smooth sensitivity $\tilde{S}_{\Delta,\beta}(G)$.

**Lemma 19.** *(Lemma 5) With probability at least $1 - \delta$ in Algorithm 2, $\max_{ij} x_{ij} \frac{\|W\|}{s} \in (1 - \theta, 1 + \theta) \max_{ij} a_{ij}^2$.*

*Proof.* We split all pairs $ij$ into 2 sets $\{i'j' : a_{i'j'}^2 < \tau\}$ and the remaining pairs $\{\tilde{i}\tilde{j}\}$ that $a_{\tilde{i}\tilde{j}}^2 > \tau$. We will prove that the Algorithm will not output $x_{i'j'}$ for any $i'j'$.

For every $i'j'$ such that $a_{i'j'}^2 < \tau$, by Lemma 16, set $\lambda = 1, \lambda' = \frac{3}{2}$, since $c \geq 12\theta^2$ or $\frac{c(\lambda - \lambda')}{\lambda \theta^2} = \frac{12\theta^2 1/4}{\theta^2} = 3$, we have $\max_{i'j'} x_{i'j'} \frac{\|W\|}{s} < \frac{3}{2}\tau$ with probability at least $1 - 1/n^{c-2}$.

Because $\max_{ij} a_{ij}^2 > \tau_k = 3\tau$, by Lemma 17, set $\lambda = 3, \lambda' = \frac{3}{2}$, since $c \geq 2$ or $c\lambda > 3$, we have $\max_{ij} x_{ij} \frac{\|W\|}{s} > \frac{3}{2}\tau$ with probability at least $1 - 1/n^{c-2}$.

Then with probability at least $1 - 1/2n^{c-2}$, Algorithm 2 will not output any $x_{i'j'}$.

The remaining sets $\{\tilde{i}\tilde{j}\}$ that $a_{\tilde{i}\tilde{j}}^2 > \tau$. By Lemma 15, $\forall \tilde{i}\tilde{j} : x_{\tilde{i}\tilde{j}} \frac{\|W\|}{s} \in (1 - \theta, 1 + \theta) a_{\tilde{i}\tilde{j}}^2$. Applying the union bound on all pair $\tilde{i}\tilde{j}$ and setting $c \geq \log_n(1/(2\delta)) + 2$, the Lemma follows. $\qquad\square$

**Theorem 5.** *(Theorem 2) $\tilde{S}_{\Delta,\beta}(G)$ output by Algorithm 2 is a $(\gamma, \delta)$-upper approxmiation to $S_{\Delta,\beta}(G)$.*

*Proof.* From the result of Lemma 19, we have:

$$(1 - \theta) \max_{ij} a_{ij}^2 \leq \max_{ij} x_{ij} \frac{\|W\|}{s} \leq (1 + \theta) \max_{ij} a_{ij}^2$$

.

Multipling all terms by $1/(1 - \theta)$, we have:

$$\max_{ij} a_{ij}^2 \leq \max_{ij} x_{ij} \frac{\|W\|}{s(1 - \theta)} \leq \frac{1 + \theta}{1 - \theta} \max_{ij} a_{ij}^2$$

$$\max_{ij} a_{ij}^2 \leq \hat{a}^2 \leq \frac{1 + \theta}{1 - \theta} \max_{ij} a_{ij}^2$$

$$\max_{ij} a_{ij} \leq \hat{a} \leq \sqrt{\frac{1 + \theta}{1 - \theta}} \max_{ij} a_{ij}$$

$$\max_{ij} a_{ij} \leq \hat{a} \leq \sqrt{e^{2\gamma}} \max_{ij} a_{ij}$$

$$\max_{ij} a_{ij} \leq \hat{a} \leq e^{\gamma} \max_{ij} a_{ij}.$$

Therefore with probability at least $1 - \delta$, we have $LS_{\Delta}^{(t)}(G) \leq \widehat{LS}_{\Delta}^{(t)}(G) \leq e^{\gamma} LS_{\Delta}^{(t)}(G)$ and hence:

$$S_{\Delta,\beta}(G) \leq \tilde{S}_{\Delta,\beta}(G) \leq e^{\gamma} S_{\Delta,\beta}(G),$$

with probability at least $1 - \delta$. It follows that $\tilde{S}_{\Delta,\beta}(G)$ is $(\gamma, \delta)$-upper approximation to $S_{\Delta,\beta}(G)$. $\qquad\square$

**Theorem 6.** *(Theorem 2) Suppose $\max_{ij} a_{ij} > 0$. Algorithm 2 has a runtime of*

$$O\left(\frac{\log_n(2\delta)\|W\|_1 \log m \log n}{\left(\frac{e^{2\gamma} - 1}{e^{2\gamma} + 1}\right)^2 \max_{ij} a_{ij}^2} + m + \frac{\log n}{\beta} + n\right).$$

*Proof.* Let $m$ denote the number of edges in $G$. Then, computing $W$ and $\|W\|$ takes $O(m)$ time and each sampling step requires $O(\log m)$ time. However, we can perform the calculation of $W$ and $\|W\|$ once and reuse them for all Algorithms (4, 1, and 2). Each sampling step in Algorithm 4 takes $O(\log m)$ time and there are $s$ iterations which results in the total cost of $O(s \log m)$.

Algorithm 1 has a run-time of $kO(s' \log m))$ where $s' = O(\|W\|_1 c \log n/(\theta^2 \tau))$ as it executes Algorithm 4 with varying $s = s'$ and $k$ is the number of search steps. Since at the end of the search $\tau_k > \max_{ij} a_{ij}^2/4$ w.h.p., we know that $k = O(\log \frac{4n^2}{\max_{ij} a_{ij}^2})$. In the first step, we start with $\tau_0 = n^2$ and reduce $\tau_k$ by a factor of $3/4$ after each step. The total running time will be $O(\|W\|_1 \log m \times \frac{c \log n}{\theta^2} \times \frac{1}{n^2}(1 + (3/4)^{-1} + \dots (3/4)^{-k}) = O\left(\frac{\|W\|_1 c \log m \log n}{\theta^2 \max_{ij} a_{ij}^2}\right)$.

Finally, Algorithm 2 has a run-time of $O\left(\frac{\log_n (2\delta)\|W\| \log m \log n}{\left(\frac{e^{2\gamma}-1}{e^{2\gamma}+1}\right)^2 \max_{ij} a_{ij}^2} + m + \frac{\log n}{\beta} + n\right)$ taking into account the time required to compute $W$ ($O(m)$) and $\tilde{S}_{\Delta,\beta}(G)$ ($O(\log(n)/\beta + n)$) and substituting the values of $c$ and $\theta$ as described in the Algorithm. $\qquad \square$

# D   Details from Section 5

As mentioned earlier, Karwa et al. [23] introduce the notion of local sensitivity of local sensitivity for private counting of k-cliques. In this work, we generalize this notion to multi-level local sensitivity. Recall that $f(G, S)$ denotes the number of embeddings of a fixed subgraph $H$ in $G$ that contains all pairs in $S$ and $f^{(k)}(G) = \max_{|S|=k} f(G, S)$. The following lemma is at the core of our approach as it relates local sensitivity to subgraph counting.

**Lemma 20.** *For all $k \geq 0$ we have $LS(f^{(k)}) \leq f^{(k+1)}(G)$.*

*Proof.* When $k = 0$ we have $LS(f^{(0)}) = LS(f) = \max_{\{i,j\} \in L} |f(G \cup (i,j)) - f(G - (i,j))| = \max_{\{i,j\} \in L} f(G, \{i,j\}) = f^{(1)}(G)$.

When $k \geq 1$, we have $LS(f^{(k)}) = \max_{\{i,j\}} |f^{(k)}(G \cup (i,j)) - f^{(k)}(G - (i,j))|$. We argue below that for each $\{i,j\} \in L$, we have $|f^{(k)}(G \cup (i,j)) - f^{(k)}(G - (i,j))| \leq f^{(k+1)}(G)$, and the statement follows. We have two cases.

**Case 1.** $(i,j) \notin G$: let $S \subset L, |S| = k$ be a subset such that $f^{(k)}(G \cup (i,j)) = f(G \cup (i,j), S)$. Let $T \subset L, |T| = k$ be a subset such that $f^{(k)}(G) = f(G, T)$. By definition of $T$, we have $f(G, S) \leq f(G, T)$. We have two sub-cases, depending on whether $\{i,j\} \notin S$ or $\{i,j\} \in S$. First, suppose $\{i,j\} \notin S$. This implies $f(G \cup (i,j), S) = f(G, S) + f(G, S \cup \{i,j\})$, and

$$\begin{aligned} |f^{(k)}(G \cup (i,j)) - f^{(k)}(G)| &= |f(G, S) + f(G, S \cup \{i,j\}) - f(G, T)| \\ &\leq f(G, S \cup \{i,j\}) \leq f^{(k+1)}(G). \end{aligned}$$

Next, suppose $\{i,j\} \in S$. Then, $f(G, S) = f(G \cup (i,j), S) \geq f(G \cup (i,j), T)$, from the definition of $S$. Since $f(\cdot, T)$ is monotone with respect to the set of edges in the graph, we have $f(G \cup (i,j), T) \geq f(G, T)$, which implies $f(G, S) \geq f(G, T)$. Combined with the earlier observation that $f(G, T) \geq f(G, S)$, we have $f(G, T) = f(G, S)$. This implies

$$|f^{(k)}(G \cup (i,j)) - f^{(k)}(G)| = |f(G \cup (i,j), S) - f(G, T)| = |f(G, S) - f(G, T)| = 0 \leq f^{(k+1)}(G)$$

Thus, when $(i,j) \notin G$, in either sub-case, we have $|f^{(k)}(G \cup (i,j)) - f^{(k)}(G)| \leq f^{(k+1)}(G)$.

**Case 2.** $(i,j) \in G$: we have $|f^{(k)}(G \cup (i,j)) - f^{(k)}(G - (i,j))| = |f^{(k)}(G) - f^{(k)}(G - (i,j))| = |f^{(k)}(G' \cup (i,j)) - f^{(k)}(G')|$, where $G' = G - (i,j)$. Repeating the argument in case 1 for the graph $G'$, we have $|f^{(k)}(G' \cup (i,j)) - f^{(k)}(G')| \leq f^{(k+1)}(G') \leq f^{(k+1)}(G)$. $\qquad \square$

**Lemma 21.** *(Lemma 4.4 of [23]) Let $\mathcal{B}$ be an $(\epsilon_1, \delta_1)$-differentially private algorithm such that $\Pr[\mathcal{B}(x) \geq LS_f(x)) > 1 - \delta_2$ for all $x$. Consider the algorithm $\mathcal{A}$ that runs $\mathcal{B}(x)$ to obtain*

*an estimate $\widetilde{LS}$ of the local sensitivity and release both $\widetilde{LS}$ and a noisy estimate $f$: $\mathcal{A}(x) = (\widetilde{LS}, f(x) + Lap(\widetilde{LS}/\epsilon_2))$, where $\widetilde{LS} = \mathcal{B}(x)$, and where $Lap(\lambda)$ is a Laplace random variable with mean 0 and scale parameter $\lambda$. Then $\mathcal{A}$ is $(\epsilon_1 + \epsilon_2, \delta_1 + e^{\epsilon_1}\delta_2)$-DP.*

---

**Algorithm 5** Estimating higher-order private local sensitivity.
**Input:** $G, \epsilon, \delta$, subgraph $H$ with $\ell$ edges
**Output:** Private estimate $g^{(0)}$ for $f(G)$

---

1: Let $k_m = \ell$
2: **for** $k = k_m - 1$ down to 0 **do**
3:    $g^{(k)}(G) = f^{(k)}(G) + g^{(k+1)}(G)\frac{\ln 1/\delta_2}{\epsilon_2} + Lap(g^{(k+1)}(G)/\epsilon_2)$
4: **end for**
5: **return** $g^{(0)}(G)$

---

We now give the proof of correctness for Algorithm 5 using Theorem 3.

**Proof of Theorem 3**

*Proof.* We prove by induction on $k \geq 1$ that $g^{(k_m - k)}(G)$ computed by Algorithm 5 is a $((k+1)\epsilon_2, \delta_2 + (k+1)e^{\epsilon_2}\delta_2)$-DP private estimate, and $\Pr[g^{(k_m - k)}(G) \geq f^{(k)}(G)] \geq 1 - \delta_2$.

The base case is $k = 1$. Note that for $k_m = \ell$, $f^{(k_m)} \leq 1$, since once $\ell$ edges are fixed, either we get a fixed subgraph, or it doesn't result in an embedding. Since $LS(f^{(k_m-1)}) \leq f^{(k_m)}$, it follows that $g^{(k_m-1)}$ is an $(\epsilon_2, 0)$-DP estimate. Further, $\Pr[g^{(k_m-1)} < f^{(k_m-1)}] = \Pr[Lap(f^{(k_m)}/\epsilon_2) < -f^{(k_m)}\ln(1/\delta_2)/\epsilon_2] < \delta_2$, which proves the base case.

Next, consider $k > 1$. From Lemma 20, $LS(f^{(k_m-k)}) \leq f^{(k_m-k+1)}$. Since $g^{(k_m-k+1)}$ is an $(k\epsilon_2, \delta_2 + ke^{\epsilon_2}\delta_2)$-DP estimate, and $\Pr[g^{(k_m-k+1)} > f^{(k_m-k+1)}] > 1 - \delta_2$ (by induction), it follows from Lemma 21 that $g^{(k_m-k)}$ is an $(k\epsilon_2 + \epsilon_2, \delta_2 + ke^{\epsilon_2}\delta_2 + e^{\epsilon_2}\delta_2)$-DP estimate. Also, $\Pr[g^{(k_m-k)} > f^{(k_m-k)}] = \Pr[Lap(g^{(k_m-k+1)}/\epsilon_2) > -g^{(k_m-k+1)}\ln 1/\delta_2/\epsilon_2] > 1 - \delta_2$, and the inductive step follows. $\square$

Finally, we analyze the running time of Algorithm 5 using Lemma 6.

**Proof of Lemma 6**

*Proof.* Observe that $f(G, S)$ can be computed in time $O(n^{2(\ell-|S|)})$, so that $f^{(k)}$ can be estimated in time $O((n^2)^\ell)$. Therefore, the total running time for Algorithm 5 is $O(n^{2\ell})$. $\square$

**Improved time for paths of length $\ell$.** We show that for paths with $\ell$ edges, the running time can be actually improved from $O(n^{2\ell})$ to $O(n^{\ell+1})$, by keeping track of slightly more complicated information. For a subset $S \subset L$ of pairs of nodes, let $h(S)$ denote the set of unique nodes in $S$. Let $\pi$ be a function, such that for each $u \in h(S)$, $\pi(u)$ specifies the position node $u$ has in the embedding of the path. Let $f(G, S, \pi)$ the number of paths containing the edges in $S$, but with the additional constraint that the ordering with respect to $\pi$ be satisfied. Then, we have $f^{(k)}(G) = \max_{|S|=k, \pi} f(G, S, \pi)$.

*Proof.* (Sketch) Observe that $f(G, S, \pi)$ can be computed in time $O(n^{\ell+1-|h(S)|})$ time, since this involves considering the nodes in the remaining positions of the path, other than those specified by $\pi$. Further, $f^{(k)}(G)$ can be computed in $O(n^{\ell+1})$ time, since we can first guess the set $h(S)$ and the ordering $\pi$ (instead of directly guessing $|S|$ pairs of nodes), and then guess the remaining nodes. $\square$

**Theorem 7.** *Let $\mathcal{B}$ be an $(\epsilon_1, \delta_1)$-differentially private algorithm such that $\Pr[\mathcal{B}(x) \geq LS(f, x)] > 1 - \delta_2$ for all $x$. Let $\hat{f}(x) \in [(1 - \epsilon_3)f(x), (1 + \epsilon_3)f(x)]$ be an approximation to $f(x)$, with $\epsilon_3$ small enough so that $\epsilon_3 f(x) \leq LS(f, x)$ for all $x$. Consider the algorithm $\mathcal{A}$ that runs $\mathcal{B}(x)$ releases: $\mathcal{A}(x) = (\widetilde{LS}, \hat{f}(x) + Lap(\widetilde{LS}/\epsilon_2))$, where $\widetilde{LS} = \mathcal{B}(x)$, and where $Lap(\lambda)$ is a Laplace random variable with mean 0 and scale parameter $\lambda$. Then $\mathcal{A}$ is $(\epsilon_1 + (3 + \epsilon_3)\epsilon_2, e^{\epsilon_1}(\delta_1 + \delta_2) + \delta_2)$-DP.*

*Proof.* The proof is a modification of Lemma 4.4 of [23]). Define the following events

$$
\begin{aligned}
\mathcal{A}(x) &= (\widetilde{LS(f,x)} = \mathcal{B}(x), \hat{f}(x) + Lap(\widetilde{LS(f,x)}/\epsilon_2)) \\
\mathcal{A}(y) &= (\widetilde{LS(f,y)} = \mathcal{B}(y), \hat{f}(y) + Lap(\widetilde{LS(f,y)}/\epsilon_2)) \\
\mathcal{A}_{mix} &= (\widetilde{LS(f,x)} = \mathcal{B}(x), \hat{f}(y) + Lap(\widetilde{LS(f,x)}/\epsilon_2))
\end{aligned}
$$

Let $P_x, P_y$ and $P_{mix}$ correspond to the distributions of these events.

We first compare $P_y(S)$ and $P_{mix}(S)$ for any subset $S$. Observe that $\frac{P_{mix}(S)}{P_y(S)} = \frac{\Pr[\mathcal{B}(x) \in S]}{\Pr[\mathcal{B}(y) \in S]}$. Since $\mathcal{B}$ is $(\epsilon_1, \delta_1)$-DP, we have $\Pr[\mathcal{B}(y) \in S] \le e^{\epsilon_1} \Pr[\mathcal{B}(x) \in S] + \delta_1$, which implies $P_y(S) \le e^{\epsilon_1} P_{mix}(S) + \delta_1$.

Next, we compare $P_{mix}(S)$ and $P_y(S)$ for any subset $S$. Let $F$ denote the event $\mathcal{B}(x) = \widetilde{LS(f,x)} \ge LS(f,x)$. Then, $\frac{P_x(S|F)}{P_{mix}(S|F)} = \frac{\Pr[Lap(z/\epsilon_2) \in S - \hat{f}(x)]}{\Pr[Lap(z/\epsilon_2) \in S - \hat{f}(y)]} \ge exp(-\frac{|\hat{f}(y) - \hat{f}(x)|}{z}\epsilon_2)$, where $z = \widetilde{LS(f,x)}$. We have that the fraction $\frac{|\hat{f}(y) - \hat{f}(x)|}{z}$

$$
\begin{aligned}
&\le \frac{1}{z} \max\{|(1+\epsilon_3)f(y) - (1-\epsilon_3)f(x)|, |(1+\epsilon_3)f(x) - (1-\epsilon_3)f(y)|\} \\
&= \frac{1}{z} \max\{|(1+\epsilon_3)(f(y) - f(x)) + 2\epsilon_3 f(x)|, |(1-\epsilon_3)(f(x) - f(y)) + 2\epsilon_3 f(x)|\} \\
&\le \frac{(1+\epsilon_3)z + 2\epsilon_3 f(x)}{z}, \text{ with probability } 1 - \delta_1 \\
&\le 3 + \epsilon_3, \text{ with probability } 1 - \delta_1.
\end{aligned}
$$

where the second inequality holds because $|f(x) - f(y)| \le LS(f,x) \le \widetilde{LS(f,x)}$, with probability $1 - \delta_1$.

This implies $\frac{P_x(S|F)}{P_{mix}(S|F)} \ge e^{-(3+\epsilon_3)\epsilon_2}$ with probability $1 - \delta_1$. Equivalently, $P_{mix}(S|F) \le e^{(3+\epsilon_3)\epsilon_2} P_x(S|F) + \delta_1$. Therefore, $P_{mix}(S) = P_{mix}(S \wedge F) + P_{mix}(S \wedge \bar{F}) \le e^{(3+\epsilon_3)\epsilon_2} P_x(S) + \delta_1 + \delta_2$. Putting everything together, $P_y(S) \le e^{\epsilon_1} P_{mix}(S) + \delta_1 \le e^{\epsilon_1 + (3+\epsilon_3)\epsilon_2} P_x(S) + e^{\epsilon_1}(\delta_1 + \delta_2) + \delta_2$. $\qquad\square$

# E   Estimating higher-order local sensitivity for path counts

Let $H$ denote paths of length $\ell$. We assume the graph $G$ has degeneracy $d$ and treewidth $t$. We show that $f^{(k)}(G) = \max_{|S|=k} f_H(G, S)$ can be computed in time $O(poly(n)(dt)^k)$ under these assumptions on $G$ (i.e., in time smaller than $O(n^{2\ell})$). Let $\sigma$ denote a degeneracy ordering of the nodes in $V$. So node $v$ has index $\sigma(v)$. Let $N_\sigma(v) = \{v' : v' \in N(v), \sigma(v') > \sigma(v)\}$ be the set of neighbors of $v$ with index larger than $\sigma(v)$; by definition of degeneracy, it follows that $|N_\sigma(v)| \le d$ for all $v$. Let $v, u$ be two nodes such that $\sigma(v) < \sigma(u)$. Then, $\texttt{dist}(v, u, \sigma, K)$ denotes the length of the shortest path from $v$ to $u$ in a graph $K$, when restricted to nodes $v'$ with $\sigma(v) < \sigma(v') < \sigma(u)$.

Let $G(\hat{E})$ denote the graph induced by $E \cup \hat{E}$. Let $N_\sigma(v, k, \hat{E}) = \{u : \texttt{dist}(v, u, \sigma, G(\hat{E})) \le k\}$ be the set of nodes within $k$-hops of $v$ in the graph $G(\hat{E})$, restricted to higher index nodes than $v$. Let $G_\sigma(v, k, \hat{E})$ denote the graph induced by the subset $N_\sigma(v, k, \hat{E})$ of nodes. We say a pair $e = (u', v')$ is an *internal anchor pair* in $G_\sigma(v, k, \hat{E})$, if $u', v' \in N_\sigma(v, k, \hat{E})$, and $(u', v') \notin E(G_\sigma(v, k, \hat{E}))$ (i.e., $(u', v')$ is a newly added edge); else, we say $(u', v')$ is an *external anchor pair* with respect to $G_\sigma(v, k, \hat{E})$. We will add internal anchor pairs sequentially. Let $e_i$ be the edge added in the $i$th step, and let $F^{(i)} = \{e_1, \ldots, e_i\}$.

**$k$-compact subgraph.** We define $(H, D)$ to be a $k$-compact subgraph if

- $H = G_\sigma(v, k, F^{(i)})$ for some node $v$ and subset $F^{(i)} = \{e_1, \ldots, e_i\}$
- $i \le k$

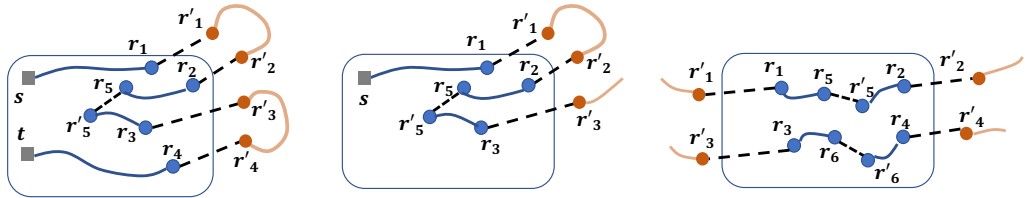

Figure 6: (Left) $D = \{r_1, r_2, r_3, r_4\}$, edges $(r_1, r_1')$, $(r_2, r_2')$, $(r_3, r_3')$ and $(r_4, r_4')$ are external anchor edges, while $(r_5, r_5')$ is an internal anchor edge. Nodes $s$ and $t$ are the starting and ending nodes of the path; (Middle) $D = \{r_1, r_2, r_3\}$, and $(r_1, r_1')$, $(r_2, r_2')$, $(r_3, r_3')$ are external anchor edges, while $(r_5, r_5')$ is an internal anchor edge. Node $s$ is the starting node of the path, and the path ends at some other node, not in this compact subgraph; (Right) $D = \{r_1, r_2, r_3, r_4\}$, edges $(r_1, r_1')$, $(r_2, r_2')$, $(r_3, r_3')$ and $(r_4, r_4')$ are external anchor edges, while $(r_5, r_5')$ and $(r_6, r_6')$ are internal anchor edges. Paths pass through this compact subgraph, without starting or ending in it.

- The pair $e_1 = (u_1, v_1)$ is an internal anchor pair in $G_\sigma(v, k, \emptyset)$. For each $j = 2, \ldots, i$, the pair $e_j = (u_j, v_j)$ is an internal anchor pair in the graph $G_\sigma(v, k, F^{(j-1)} = \{e_1, \ldots, e_{j-1}\})$.
- $D$ is a subset of vertices in $H$, and is referred to as the set of connectors.

**Lemma 22.** *Let $(H, D)$ be a $k$-compact subgraph. Then $H$ has $O(kd^k)$ nodes and edges. and the tuple $(H, D)$ can be described using $O(k^2 d^{k+1} \log d)$ bits.*

*Proof.* (Sketch) We prove by induction on $i$ that $G_\sigma(v, k, F^{(i)})$ has at most $(i + 1)d^k$ nodes and edges.

First observe that $N_\sigma(v, k, \emptyset)$ has at most $d^k$ nodes and edges, since that all have index more than $\sigma(v)$. We have $i = 1$ as the base case. Let $\sigma(u_1) < \sigma(v_1)$. Then $G_\sigma(v, k, \{e_1\})$ will have at most $2d^k$ more nodes, with at most $d^k$ additional reachable nodes through $v_1$. In the inductive step, $G_\sigma(v, k, F^{(i)} = \{e_1, \ldots, e_i\})$ has at most $d^k$ nodes (through $v_i$, assuming $\sigma(u_i) < \sigma(v_i)$) in addition to those in $G_\sigma(v, k, F^{(i-1)})$.

The neighborhood of each node $u \in N_\sigma(v, k, F^{(i)})$ can be represented in $O(d \log(kd^k))$ bits. Therefore, the total number of bits needed to specify $H$ is $O(kd^k d \log(kd^k)) = O(kd^{k+1} \log(kd^k))$. $D$ consists of at most $k$ nodes from $N_\sigma(v, k, F^{(i)})$; therefore, $D$ can be specified in $O(k \log(kd^k))$ bits. Therefore, $(H, D)$ can be specified in $O(kd^{k+1} \log(kd^k))$ bits, which implies the number of distinct $(H, D)$ pairs is at most $exp(kd^{k+1} \log(kd^k))$. $\qquad\square$

**Pairing of connector set.** The connector set $D = \{r_1, \ldots, r_\ell\}$, with $\ell \leq k$, represents nodes which have incident edges outside the compact subgraph $H$. We might have to consider three kinds of paths, corresponding to Figure 6

1. As in Figure 6 (Left), these start at a node $s$ in $H$, then pass through multiple anchor edges, and end at a node $t$ in $H$. We partition $D$ into two singletons and multiple pairs $\pi_2(D) = \{(r_{\pi(1)}), (r_{\pi(2)}, r_{\pi(3)}), \ldots, (r_{\pi(\ell)})\}$. This corresponds to paths which leave $H$ at node $r_{\pi(1)}$ and finally return at node $r_{\pi(\ell)}$, with disjoint segments from $r_{\pi(2j)}$ to $r_{\pi(2j+1)}$ lying within $H$, for each $j$. We refer to the set of such pairings as $\Pi_2(D)$.

2. This corresponds to paths which start at a node $s$ in $H$, then pass through multiple anchor edges, and don't return to a node in $H$ eventually (as in Figure 6 (Middle)). We partition $D$ into one singleton and multiple pairs $\pi_1(D) = \{(r_{\pi(1)}), (r_{\pi(2)}, r_{\pi(3)}), \ldots, (r_{\pi(\ell-1)}, r_{\pi(\ell)})\}$. We refer to the set of such pairings as $\Pi_1(D)$.

3. This corresponds to paths which only pass through nodes in $H$ (as in Figure 6 (Right)). We partition $D$ into multiple pairs $\pi_0(D) = \{(r_{\pi(1)}, r_{\pi(2)}), \ldots, (r_{\pi(\ell-1)}, r_{\pi(\ell)})\}$. We refer to the set of such pairings as $\Pi_0(D)$.

**Approximate counts corresponding to pairings.** Note that for a given compact subgraph $(H, D)$, there can be multiple pairings in $\Pi_0(D), \Pi_1(D), \Pi_2(D)$. For each such pairing, e.g., $\pi \in \Pi_0(D) = \{(r_{\pi(1)}, r_{\pi(2)}), \ldots, (r_{\pi(\ell-1)}, r_{\pi(\ell)})\}$, we can guess a set of tuples of numbers $\text{Num}(\pi, \hat{E}) = \{((\text{Num}(r_{\pi(2j+1)}, r_{\pi(2j+2)}), j = 0, 1 \ldots))\}$, where

- $\text{Num}(r_{\pi(2j+1)}, r_{\pi(2j+2)})$ corresponds to the number of segments from $r_{\pi(2j+1)}$ to $r_{\pi(2j+2)}$, which are within $H$, and use some edges from $\hat{E}$ as internal anchor edges

- The path segments counted in $\text{Num}(r_{\pi(2j+1)}, r_{\pi(2j+2)})$ and those in $\text{Num}(r_{\pi(2j'+1)}, r_{\pi(2j'+2)})$ are disjoint. In particular, this means that the internal anchor edges used in these counts are disjoint.

- The paths within a tuple $((\text{Num}(r_{\pi(2j+1)}, r_{\pi(2j+2)}), j = 0, 1 \ldots))$ contain all the internal anchor edges in $\hat{E}$.

In order to reduce the information complexity, we only keep track of $\text{Num}(r_{\pi(2j+1)}, r_{\pi(2j+2)})$ approximately as powers of $(1 + \mu)$ for a small $\mu \in (0, 1)$; we denote these approximate counts as $\widehat{\text{Num}}(r_{\pi(2j+1)}, r_{\pi(2j+2)})$, and the corresponding tuple of counts by $(\widehat{\text{Num}}(r_{\pi(1)}, r_{\pi(2)}), \ldots, \widehat{\text{Num}}(r_{\pi(\ell-1)}, r_{\pi(\ell)}))$. $\widehat{\text{Num}}(\pi, \hat{E})$ denotes the set of approximate path counts which use the internal anchor edges $\hat{E}$. Similarly, approximate counts $\widehat{\text{Num}}(\pi, \hat{E})$ can be defined for $\pi \in \Pi_1(D)$ and $\pi \in \Pi_2(D)$. Let $\Pi(D) = \Pi_0(D) \cup \Pi_1(D) \cup \Pi_2(D)$.

Let $IA(H)$ denote the set of internal anchor edges in a compact subgraph $H$. We refer to a tuple $(H, D, \Pi(D), \bigcup_{\pi \in \Pi(D)} \widehat{\text{Num}}(\pi, IA(H)))$ as an *augmented compact subgraph with approximate counts*.

**Feasible solutions as a tuple of augmented compact subgraph with approximate counts.** We now observe that any feasible solution involving anchoring a subset $S = \{(i_r, j_r) : r = 1, \ldots, s\}$ of pairs of nodes can be viewed as a set of at most $k$ disjoint augmented compact subgraphs with approximate counts.

**Lemma 23.** *Any feasible solution* $S = \{(i_r, j_r) : r = 1, \ldots, s\}$, *with* $|S| \leq k$ *corresponds to a set of augmented compact subgraphs with approximate counts* $(H_1, D_1, \Pi(D_1), \bigcup_{\pi \in \Pi(D_1)} \widehat{\text{Num}}(\pi, IA(H_1))), \ldots, (H_\ell, D_\ell, \Pi(D_\ell), \bigcup_{\pi \in \Pi(D_\ell)} \widehat{\text{Num}}(\pi, IA(H_\ell)))$, *for some* $\ell \leq k$, *such that for every* $j$, $(H_j, D_j)$ *consists of nodes with lower index than those in* $(H_{j'}, D_{j'})$, *for* $j < j'$.

*Proof.* (Sketch) Consider the set of all paths $\mathcal{P}$ which are considered, i.e., those which use the edges from $S$. Let $v$ be the node with the minimum index among those in $\mathcal{P}$. Consider the graph $G_\sigma(v, k, \emptyset)$. If any pair $(i_s, j_s) \in S$ is an internal anchor pair for $G_\sigma(v, k, \emptyset)$, define $e_1 = (i_s, j_s)$ (this is added to $IA(H_1)$); this is again repeated if there is another pair $(i_{s'}, j_{s'})$ which is an internal anchor pair for $G_\sigma(v, k, e_1)$. We stop, when we have a subgraph $H_1$ such that no other pair in $S$ is an internal anchor pair for $H_1$. If there is a path $P \in \mathcal{P}$ with one end point $u \in V(H_1)$ and other end point outside $H_1$, $u$ will be added to $D_1$. Finally, we construct pairings and associated approximate counts. If a path $P$ has a segment $P'$ from $r_1 \in D_1$ to $r_2 \in D_1$ which is within $H_1$, we would have $(r_1, r_2)$ as part of a pairing, and the number of such segments will contribute to the associated counts. Thus a compact subgraph $(H_1, D_1, \Pi(D_1), \bigcup_{\pi \in \Pi(D_1)} \widehat{\text{Num}}(\pi, IA(H_1)))$ can be constructed corresponding to the set $\mathcal{P}$.

Next, consider the path segments we get once we delete all the segments in $H_1$. The next compact subgraph is constructed in a similar manner.

$\square$

Let $\mathcal{S}(G, \sigma, k)$ denote the set of possible augmented compact subgraphs with approximate counts. From Lemma 23, it follows that any feasible solution is an element of $\bigcup_{k' \leq k} \mathcal{S}(G, \sigma, k)^{k'}$.

**Lemma 24.** *The number of feasible solutions is* $O(\exp(k^k d^{k+1} \log(kd^k))(k \log n)^{k^2})$.

*Proof.* (Sketch) We first bound $|\mathcal{S}(G, \sigma, k)|$. For each compact subgraph $(H, D)$, the number of pairings $|\Pi(D)|$ is at most $k^k$. For each pairing $\pi \in \Pi(D)$, there are $O(\log n)^k$ possible approximate

counts. Therefore, $|\mathcal{S}(G, \sigma, k)| = O(exp(kd^{k+1} \log(kd^k))(k \log n)^k$ using Lemma 22. The lemma follows since the number of feasible solutions is at most $k|\mathcal{S}(G, \sigma, k)|^k$. $\qquad\square$

**Finding a tuple of disjoint augmented compact subgraphs with approximate counts in $G$.** We now show that we can check in polynomial time if an augmented compact subgraph exists. Thus in time $O(|\mathcal{S}(G, \sigma, k)|poly(n))$, we can check which augmented compact subgraphs exist. For a given augmented compact subgraph, the total number of paths can be approximated, which allows us to find the one with the maximum number.

**Lemma 25.** *Suppose we are given a tuple $M_1, \ldots, M_\ell$, where $M_j = (H_j, D_j, \Pi(D_j), \{\widehat{Num}(\pi), \pi \in \Pi(D_j)\})$, for some $\ell \le k$ such that for every $j$, $(H_j, D_j)$ consists of nodes with lower index than those in $(H_{j'}, D_{j'})$, for $j < j'$. It is possible to find a set of nodes $v_1, \ldots, v_\ell$, sequence of edges $F^{(1)}, \ldots, F^{(\ell)}$, connector sets $\hat{D}_1, \ldots, \hat{D}_\ell$, pairings $\Pi(\hat{D}_1), \ldots, \Pi(\hat{D}_\ell)$, such that for each $j$, $H_j = G_\sigma(v_j, k, F^{(j)})$, $\hat{D}_j \subset V(G_\sigma(v_j, k, F^{(j)}))$, and $\bigcup_{\pi \in \Pi(\hat{D}_j)} \widehat{Num}(\pi, IA(H_j) = \{\widehat{Num}(\pi), \pi \in \Pi(D_j)\}$ in time $O(poly(n)exp(k^k d^{k+1} \log(kd^k))(k \log n)^{k^2}(kd^k)^t)$ if $G$ has degeneracy $d$ and treewidth $t$.*

*Proof.* (Sketch) We use $c_j$ to denote a color corresponding to $M_j$. For each node $v$, it is possible to decide in time $O(kd^k)$ if $v$ can be colored by $c_j$, i.e., if there exist $F^{(j)}, \hat{D}_j$ such that $(G_\sigma(v, k, F^{(j)}), \hat{D}_j, \bigcup_{\pi \in \Pi(\hat{D}_j)} \widehat{Num}(\pi, IA(H_j))) = M_j$. If $v$ has color $c_j$, it has "conflicts" with $C(v, c_j) = V(G_\sigma(v, k, F^{(j)}))$, i.e., we cannot have a node $u \in C(v, c_j)$ assigned color $c_i$. Our goal is to find nodes $v_1, \ldots, v_\ell$ such that $v_j$ can be assigned color $c_j$, and $v_{j'} \notin C(v_j, c_j)$ for any $j' \ne j$. If $G$ has treewidth $t$, we can determine if such a coloring can be done in time $O(poly(n)(kd^k)^t)$ time. The main idea for the dynamic program is the following.

Let $\mathcal{T} = (\mathcal{V}, \mathcal{E})$ denote a tree decomposition of $G$. Each "node" $U \in \mathcal{V}$ is a subset of $V$. We construct an information set $I(U)$ for each $U \in \mathcal{V}$, which will allow us to determine possible colors for each node $v \in U$, in the following manner. $I(U)$ consists of a set of possible colorings $\mathbf{x}$ of $U$. We consider only $\mathbf{x}$ which correspond to colorings in which each color appears at most once; $\mathbf{x}$ can be viewed as a coloring of the subgraph induced by all the nodes in the subtree of $\mathcal{T}$ rooted at $U$. A coloring $\mathbf{x}$ specifies for each node $u \in U$, and for each $j$, an indication of the following: (1) node $u$ is colored by $c_j$, (2) node $u$ is not colored by $c_j$, but color $c_j$ that can be assigned to it, (3) node $u$ is not colored by $c_j$, and, moreover, $c_j$ cannot be assigned to $u$. This is because there exists some other node $u'$ such that $u \in C(u', c_j)$. We don't keep track of node $u'$ (otherwise we would end up needing $n^k$ information), but instead specify the set $F$ and position of $u$ in the compact graph $G_\sigma(u', k, F)$. The information set $I(U)$ consists of all possible such colorings $\mathbf{x}$; note that $|I(U)| = O((kd^k)^t)$.

We construct the information sets of all $U \in \mathcal{V}$ in a bottom-up manner. Suppose $U$ has children $U_1, U_2 \in \mathcal{V}$. We construct $I(U)$ from $I(U_1)$ and $I(U_2)$, as described below. For each $\mathbf{x}' \in I(U_1), \mathbf{x}'' \in I(U_2)$, we construct a coloring $\mathbf{x}$, if feasible, in the following manner. For each node $u \in U$, and for each $j$:

1. If $u \in U_1 \cap U_2$, $\mathbf{x}$ is infeasible if the indicators for $u$ in $\mathbf{x}'$ and $\mathbf{x}''$ are inconsistent

2. If $u$ is contained only in $U_1$, the indicator for $u$ in $\mathbf{x}$ is kept to be the same as that in $\mathbf{x}'$; similarly if $u$ is contained only in $U_2$

3. If $u \notin U_1 \cup U_2$, we check if $N(u) \cap (U_1 \cup U_2) \ne \emptyset$. In this case, we choose an indicator for $u$ in $\mathbf{x}$ that is consistent with the indicators for its neighbors in $\mathbf{x}'$ and $\mathbf{x}''$. It is also possible that we have inconsistencies, and in that case, $\mathbf{x}$ will not be feasible. If a node $u_1 \in N(u) \cap U_1$ is in the conflict set of some node with color $c_j$, using the position of $u_1$ in the associated compact subgraph, we can determine if $u$ would also have a conflict, and if so, at what position. Similarly, if there exists a node $u_2 \in N(u) \cap U_2$. If there simultaneously exist $u_1 \in N(u) \cap U_1, u_2 \in N(u) \cap U_2$, both of which are in conflict sets with respect to $c_j$, which induce inconsistent conflicts for node $u$ with respect to $c_j$, we would consider $\mathbf{x}$ infeasible; if the conflicts induced by $u_1, u_2$ are consistent, we could keep that indication for node $u$ with respect to $c_j$.

Once the information sets $I(U)$ for all $U \in \mathcal{V}$ are known, we can find a valid coloring in a top-down manner. □

**Lemma 26.** *Suppose we are given a tuple $M_1, \ldots, M_\ell$, where $M_j = (H_j, D_j, \Pi(D_j), \{\widehat{Num}(\pi), \pi \in \Pi(D_j)\})$, for some $\ell \leq k$ such that for every $j$, $(H_j, D_j)$ consists of nodes with lower index than those in $(H_{j'}, D_{j'})$, for $j < j'$. It is possible to determine in $O(k^k)$ time if the $M_j$'s are consistent, i.e., if it is possible to find external anchor edges that are consistent for the tuples.*

**Theorem 8.** *If $G$ has degeneracy $d$ and treewidth $t$, we can get a $O(1 + 1/polylog(n))$ factor upper bound on $f(G, S)$ in time $O(poly(n)exp(k^k d^{k+1} \log(kd^k))(k \log n)^{k^2} (kd^k)^t)$, .*

## F   Additional Experiments

In this section, we present the complete set of experimental results for private triangle counting (Algorithm 2) on all the tested networks (12 networks given in Table 2).

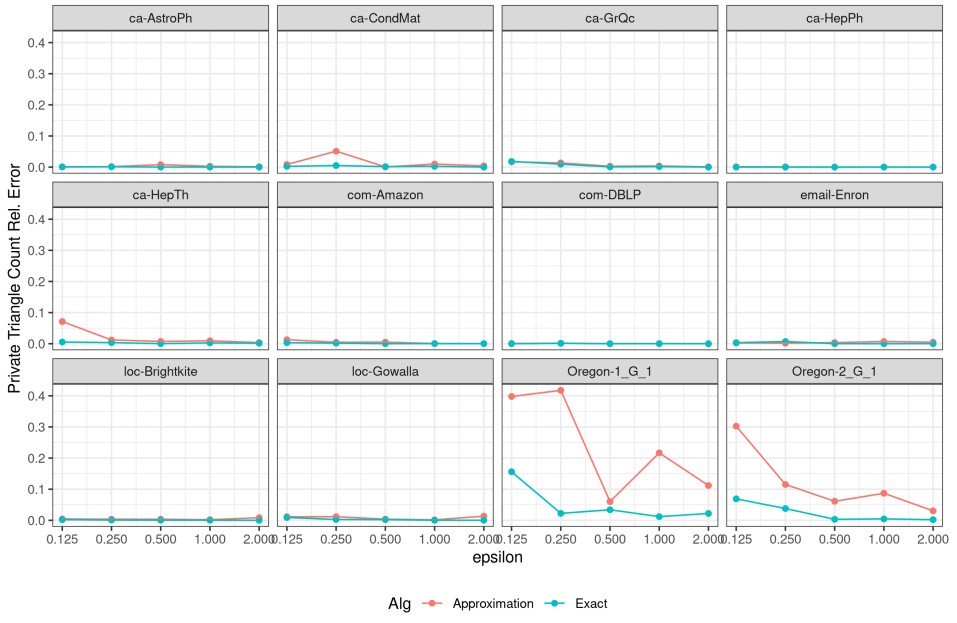

Figure 7: Triangle Count Relative Error, showing the accuracy of the private triangle count with noise calibrated by (1) approximate smooth sensitivity (Algorithm 2), and (2) exact smooth sensitivity.

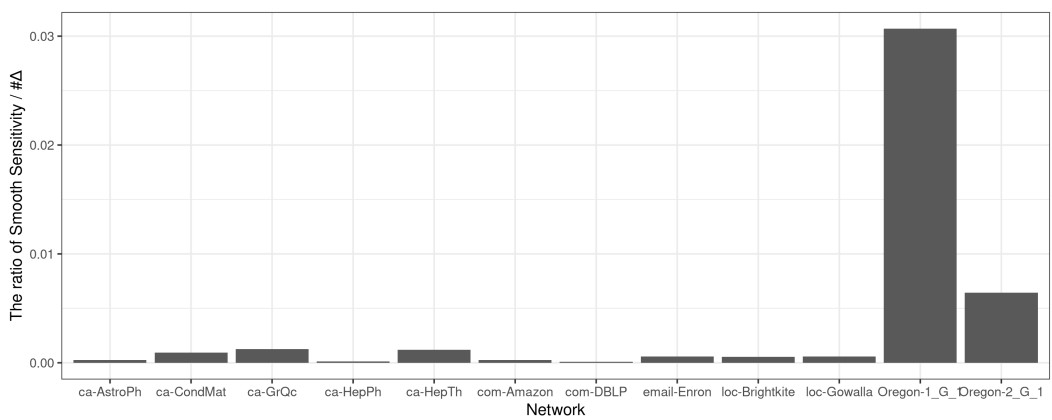

Figure 8: The ratio of the smooth sensitivity over the actual count of triangles for all networks.

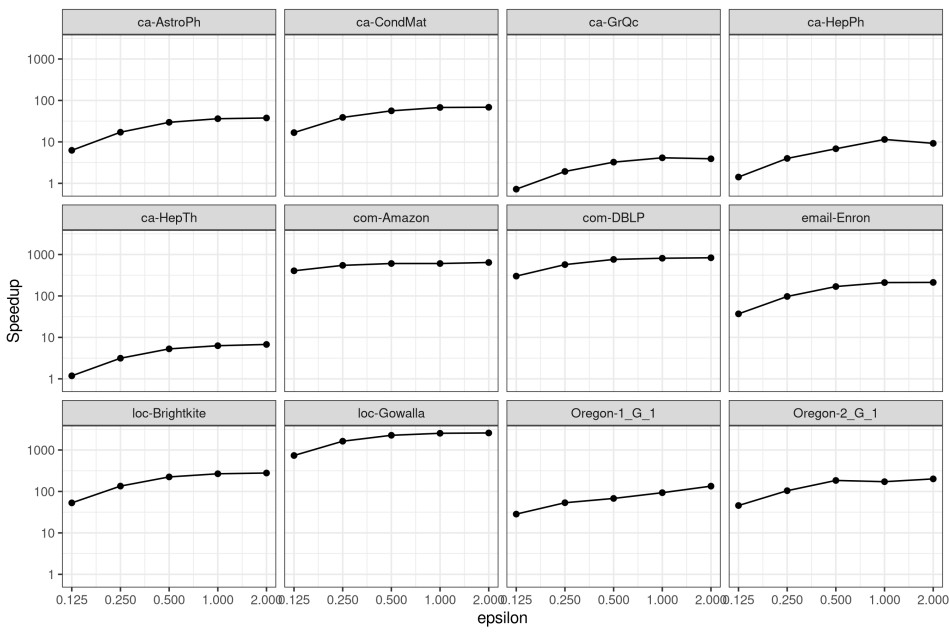

Figure 9: Speedup of Algorithm 2 (compared to the exact calculation) on all networks.

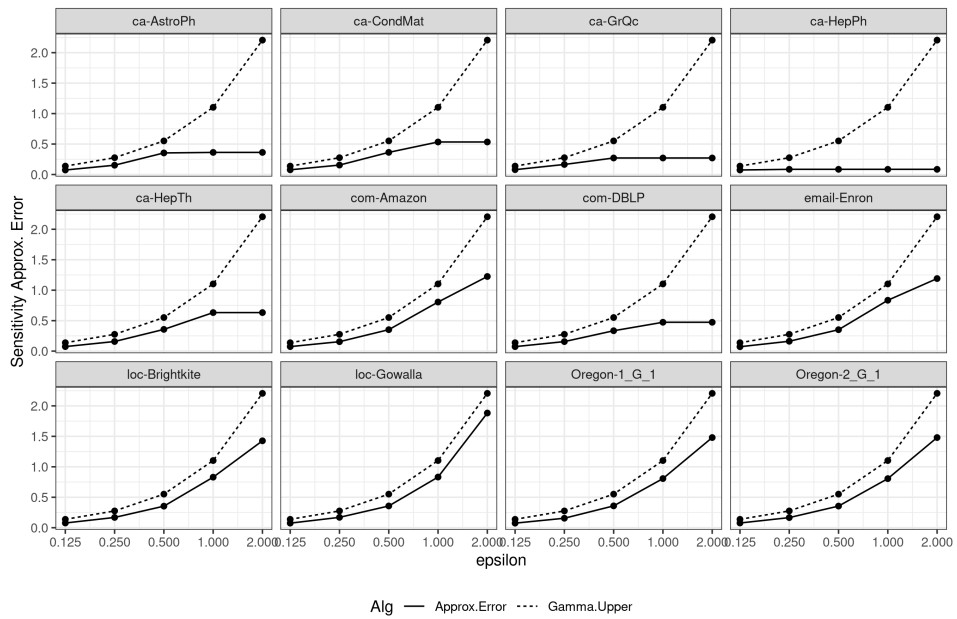

Figure 10: Sensitivity Approx. Error, showing the error of approximate smooth sensitivity output by Algorithm 2 (relative to the exact smooth sensitivity) on all networks.

Figure F shows that when approximate smooth sensitivity is used instead of exact smooth sensitivity, the accuracy of the private triangle counts measured using the metric, Triangle Count Relative Error, are the same across different privacy budgets in all but two networks (Oregon 1 and Oregon 2) (Figure 8). In these two networks, the smooth sensitivities are relatively large in comparison with the triangle counts, making the noise fluctuate much more for the approximate estimate. For the ca-HepPH dataset, the two outputs coincide.

Figure 8) shows the ratio of smooth sensitivity to actual triangle count for all datasets. Note that this ratio is large for the two datasets, Oregon 1 and 2, but small for all the other datasets.

Figure 9 shows the speed-up achieved using Algorithm 2. As seen in the figure, it is orders of magnitude faster than the exact algorithm on all the 12 datasets. In fact, on large graphs (Amazon, DBLP, Gowalla), the speedup is as high as $1,000$-fold. Generally, a lower $\epsilon$ requires a smaller approximation factor ($\gamma$), which in turn requires a larger number of iterations to reduce the error in approximation. To summarize, the majority of tested networks have speedup factors between 10 and 100-fold across all privacy budgets $\epsilon$.

Figure 10 shows that all approximate smooth sensitivities computed by Algorithm 2 are close to the exact smooth sensitivities. We measure this using the metric, Sensitivity Approx. Error. Approximate smooth sensitivities are always larger than the true smooth sensitivity. It is necessary since a lower value may expose the privacy due to an inadequate noise magnitude. In general, a smaller value of $\epsilon$ (higher privacy guarantee) requires a more accurate estimation of the sensitivity. It is illustrated in Figure 10 as the approximate smooth sensitivity is close to the exact smooth sensitivity in all networks at $\epsilon = 0.125$ or $\epsilon = 0.25$.

