# OpenReview forum: "Faster approximate subgraph counts with privacy"
_NeurIPS.cc/2023/Conference — NeurIPS 2023 poster_

### Official Review · Reviewer_LedL · 2023-06-18

**Soundness:** 4 excellent
**Presentation:** 4 excellent
**Contribution:** 3 good
**Rating:** 6
**Confidence:** 4

**Summary:**

The paper considers the problem of releasing information about graphs (specifically, subgraph counts) under differential privacy.  This has been a widely studied problem, due to the challenging nature of ensuring that the noise added is bounded.  Many techniques (e.g., smooth sensitivity) have been developed within the privacy community motivated by such graph problems.  A particular concern is the computational cost: many techniques have a high polynomial or exponential cost, rendering them impractical -- and more costly than their non-private counterparts.  The present paper studies how approximation can adopted, in two ways: applying approximation to the sensitivity calculations, and to the quantity of interest.  In some ways, it is perhaps not surprising that bounds can be obtained by tolerating approximation, a common technique in randomized algorithms.  Recent work has started to look at the intersection of approximation and privacy (e.g., in the case of quantile computations, https://arxiv.org/abs/2201.03380), so it is natural to apply it here.  Specific results are shown for triangles, paths and trees, as subgraphs of particular interest.

**Strengths:**

These are fundamental questions about the effect of privacy -- motivated by practical concerns, but perhaps contributing more to out theoretical understanding of whether we can obtain private counterparts of exact or approximate algorithms for subgraph counting.  The paper opens up the question of using approximation in different ways in the private algorithm design, and focuses attention on the computational cost of the algorithms, whereas prior work has looked primarily at the accuracy.

The technical details (in full in the supplemental material) is detailed and challenging in places, although making use of standard probability tools (e.g., Chernoff bounds).

**Weaknesses:**

The paper takes great pains to cite recent work by Blocki et al [4] which as far as I can tell is currently an unpublished technical report.  It summarizes the difference as studying approximate sensitivity.  Still, I think it would be useful to provide more contrast.  For instance, the empirical study compares against the original work on smooth sensitivity from 2007.  Many papers relevant to private triangle counting have come since then.  In particular, it would be instructive to also compare against the black-box approach of Blocki et al to see where it falls on these axes.

**Questions:**

Lemma 2 seems like it should be obvious: if we have an estimate of the smooth sensitivity that is always bigger except with probability delta', then we should be able to directly use this to add noise based on the estimate, and only incur delta' additional "failure probability" in the (eps, delta)-differential privacy bounds.  But the proof in the supplementary material works through a longer argument adapting the original proof of smooth sensitivity.  Can you comment on whether the intuitive argument above is valid.

In Defn 5, wouldn't it be quicker to say that g_f is A_f(D) clipped to the range [ (1-alpha)f(D), (1+alpha)f(D) ]?  And similarly for s_f.

Theorem 2, the dependence should be on log(1/(2 delta)) ?

**Limitations:**

No limitations around societal impact.
The paper could discuss more the limitations of the approach and future work -- Section 7 is very brief on this topic.

---

> ### Author Rebuttal · Authors · 2023-08-09
>
> W1. The paper takes great pains to cite recent work by Blocki et al [4] which as far as I can tell is currently an unpublished technical report. It summarizes the difference as studying approximate sensitivity. Still, I think it would be useful to provide more contrast. For instance, the empirical study compares against the original work on smooth sensitivity from 2007. Many papers relevant to private triangle counting have come since then. In particular, it would be instructive to also compare against the black-box approach of Blocki et al to see where it falls on these axes.
>
> *Response.* To the best of our knowledge, and as per section 3.2.1.1 of the survey by (Li et al., ACM Computing Surveys, 2023), there are very few recent papers which have considered private subgraph counting problems, or even triangle counting in the edge differential privacy model.
> The survey by Li et al. mentions two other papers relevant to triangle counting, in addition to the ones we cite in our paper.
> The first, (Chen et al., SIGMOD 2013) develops a recursive strategy.
> However, as noted in the criticism of this work in (Zhang et al., 2015), their methods ``release a lower bound on the result, whose global sensitivity is relatively low. This method suffers from a bias between the
> true query answer and the lower bound, in exchange for less noise''.
> The second paper by (Proserpio et al., 2014) presents a different approach which estimates counts by degree, which can be implemented in time $O((\max_v deg(v))^3 m)$.
> We will discuss all these papers, but note that our result improves on all of them.
> There are some other papers on subgraph counting (including triangle counting) under the node differential privacy model, e.g., (Kasiviswanathan, et al., 2013) and (Ding et al., 2021).
> Similarly, there are ither works, such as (Imola et al., 2022) on subgraph counting in the shuffle model.
> We haven't considered a comparison with these results, because the privacy model is different.
>
> Q1. Lemma 2 seems like it should be obvious: if we have an estimate of the smooth sensitivity that is always bigger except with probability delta', then we should be able to directly use this to add noise based on the estimate, and only incur delta' additional "failure probability" in the (eps, delta)-differential privacy bounds. But the proof in the supplementary material works through a longer argument adapting the original proof of smooth sensitivity. Can you comment on whether the intuitive argument above is valid.
>
> *Response.* Actually, this argument will fail the Dilation Property (at the end of Lemma 10): $S_{f,\beta} (D)/S_{f,\beta} (D')$ will be unbounded.
>  Formally, let $\tilde{S}$ be an approximation of the $\beta$-smooth sensitivity $S^*$ such that $\tilde{S}(D) > S^*(D)$ with probability $1 - \delta$ for any dataset $D$. We can see that the Dilation Property (at the end of Lemma 10) does not go through, since the ratio $\frac{\tilde{S}(D)}{\tilde{S}(D')}$ is unbounded, hence we cannot use $\tilde{S}$ for noise calibration.
>
> Q2. In Defn 5, wouldn't it be quicker to say that $g_f$ is $A_f(D)$ clipped to the range $[ (1-alpha)f(D), (1+alpha)f(D) ]$? And similarly for $s_f$.
>
> *Response.* Yes, that is correct.
> We used this notation to keep it consistent with Blocki et al., but will clarify this in the paper.
>
> Q3. Theorem 2, the dependence should be on $\log(1/(2 \delta))$ ?
>
> *Response.* Yes. It is a good catch, and we will fix this.
> The numerator involves a factor of c, which is set to $\log (1/(2\delta))$ in Algorithm 2.

---

> > ### Comment · Reviewer_LedL · 2023-08-11
> >
> > I thank the authors for their careful response, and the clarification around using bounds for the smooth sensitivity.
> >
> > Regarding the coverage of prior work, I think it is not sufficient to discount the later works in edge DP on the grounds that they have some perceived weaknesses.  The paper would be more compelling if it included empirical evidence about their performance.  I am also unsure that it is OK to ignore work in the (stronger) node differential privacy model or shuffle model: it is possible that the techniques here could be easily adapted to the edge DP model, so it would strengthen this paper to give greater study to these papers.

---

> > > ### Author Response · Authors · 2023-08-17
> > >
> > > We thank the reviewer for their comments and suggestions.
> > > Please see our response below on three points.
> > >
> > > **1. Extensions of our methods to node DP models.**
> > >
> > > After the reviewer's suggestion, we realized that some of our results would extend to node DP.
> > > Theorem 1 in our paper on the analysis of approximate smooth sensitivity holds for node DP as well, and we expect this could be used for improved private queries for other problems.
> > > However, the specific technique using diamond sampling for faster triangle counting only holds in the edge DP model.
> > > Fundamentally new ideas are needed for extending this technique to node DP and other models.
> > > Our result based on higher-order local sensitivity (Lemma 6) also extends to node DP.
> > > It is possible that our improved analysis for path counting in special graphs (Theorem 8) works for node DP, though we haven't fully worked this out.
> > > We will write these in the final paper.
> > >
> > > **2. Experimental comparison with other methods**
> > >
> > > We implemented the Ladder function (Zhang et al. 2015. ref[33]) and evaluated it using the same settings and metrics we use in our Experiments section. We set up the experiments for some small-medium size networks (20K-3M triangles). We report and visualize here (*the anonymized link to the figures was sent to the AC per Neurips guideline*) the accuracy metric between Our algorithm, Karwa et al. , and the Ladder function (Zhang et al.). In general, we can see that the Karwa et al. and Zhang et al. have similar accuracy, with the Zhang et al. having a slight advantage on smaller $\epsilon$. Our algorithm can match their performance in many settings, especially for $\epsilon$ larger than 1.
> > > Note that some lines overlap in HepTh and HepPh datasets due to very similar performance among all 3 methods. **For the runtime, we report the Speedup of our algorithm over the other methods. It shows that our algorithms are orders of magnitude faster than both of them.** We will conduct and report the complete experiments on our final version. Note that in (Zhang et al.)'s experiment, the recursive mechanism (Chen et al.) has a similar utility as the Ladder function for triangle counting task, but is designed and implemented as join queries of relational DBMS and it is unclear to us how to create a fair performance comparison.
> > >
> > > We would like to point out that the (Blocki et al., 2023) paper has no experimental results.
> > > This makes it difficult for us to compare our results with their methods directly.
> > > We also note that our approximate smooth sensitivity and Karwa et al. vs (Blocki et al. 2023) focus on different contexts. Ours and (Karwa et al.) discuss the utility and performance of the computation of the alternatives for global sensitivity of counting triangles, with the assumption that we have the exact count of triangles ready (so no computation here). (Blocki et al. 2023) discuss a black box method for "approximate triangle counting algorithm" that it has to estimate the number of triangles inside the blackbox and then adding a constant-like magintude noise (without further computation, the noise magnitude depends on the triangle estimation). In other words, Ours and (Karwa et al.) compute/approximate the noises and treat the triangle count as known constant, while (Blocki et al. 2023) takes the noises as some constant-like and approximate the triangle counts.
> > >
> > > **3. Privacy models used in prior papers.**
> > > We note that the initial papers on graph DP, which developed conceptual tools for private analysis, e.g., smooth sensitivity, mentioned relevance to multiple privacy models.
> > > However, specific results were almost always for a single privacy model, as we summarize below.
> > >
> > > 1. (Nissim et al., 2007), ref [29]: develop the original smooth sensitivity result, and show applications to triangle counting (edge DP) and MST (weight DP, which is a slight generalization of edge DP)
> > > 2. (Blocki et al. 2012), ref [3]: considers both edge and node DP, but uses two different methods for these two models
> > > 3. (Cohen-Addad et al., 2022), ref [8]: correlation clustering and other problems for edge DP
> > > 4. (Dhulipala et al., 2022), ref [9]: edge DP
> > > 5. (Imola et al., 2021), ref [15], and (Imola et al., 2022), ref [16]: edge local DP
> > > 6. (Karwa et al., 2014), ref [21]: edge DP
> > > 7. (Kasiviswanathan et al., 2013), ref [23]: node DP
> > > 8. (Nguyen et al., 2021), ref [27]: edge DP
> > > 9. (Nguyen et al., 2016), ref [28]: edge DP
> > > 10. (Zhang et al., 2015), ref [33]: edge DP
> > >
> > > In summary, most papers present results for graph problems in a specific DP model.
> > > Therefore, our focus on edge DP is not really a limitation, and the fact that some of our results hold for both edge and node DP is an additional strength.

---

> > > > ### Comment · Reviewer_LedL · 2023-08-18
> > > >
> > > > Thank you for this detailed response, and the new observations that some results will extend to node DP.  I agree with reviewer Bhh2that the paper can be significantly enhanced by incorporating this more thorough categorization and comparison to past work.  Other clarifications highlighted in the parallel discussion threads will also improve the impact of the work.  Highlighting the extent to which techniques do or do not transfer across different privacy models will be a useful contribution for the community.

---

> > > > > ### Author Response · Authors · 2023-08-20
> > > > >
> > > > > We thank the reviewer for their positive feedback and support. In the final version, we will ensure that all the details regarding the extension of our results to the node DP model, comparison to previous works (including our new experimental results for the ladder function method), and our clarifications in other threads of the rebuttal are clearly highlighted.

---

### Official Review · Reviewer_Bhh2 · 2023-07-03

**Soundness:** 3 good
**Presentation:** 3 good
**Contribution:** 3 good
**Rating:** 6
**Confidence:** 2

**Summary:**

The paper focuses on the problem of counting the number of non-induced embeddings of a subgraph while ensuring differential privacy. The objective is to develop efficient DP algorithms for subgraph counting that outperform previous approaches in terms of running time.

To accomplish this goal, the authors give a method that leverages approximate smooth sensitivity as a substitute for global sensitivity in privacy mechanisms. This approach proves effective in accurately estimating smooth sensitivity for tasks such as triangle counting and path counting.

In addition, the paper presents experimental results for triangle counting using real-world data. These experiments demonstrate that the algorithm's estimation outcomes closely align with the accurate values.

**Strengths:**

The problem of counting the number of subgraphs is an important question in the graph differential privacy community.

The idea of using approximate smooth sensitivity to improve the running times of current algorithms is somehow natural and interesting.


**Weaknesses:**

The running time of the DP algorithm for (arbitrary) subgraphs with $\ell$ edges is $O(n^{2\ell})$, which is still quite inefficient.


**Questions:**

For the DP algorithms given in the paper, the output A(D) is an approximation of the quantity that we want to compute (e.g., the number of triangles). But the approximation ratio of A(D) is never mentioned, i.e. what is the multiplicative error and what is the additive error? Furthermore, how could you compare these algorithms to previous algorithms in terms of approximation ratios?

What is the state-of-the-art of running times of the DP algorithms for counting subgraphs with ell edges?

Why the experiments are only for triangles? What about paths or other subgraphs?

**Limitations:**

See "Weaknesses".

---

> ### Author Rebuttal · Authors · 2023-08-09
>
> W1. The running time of the DP algorithm for (arbitrary) subgraphs with edges is $O(n^{2l})$, which is still quite inefficient.
>
> *Response.* We completely agree with the reviewer's assessment about the practicality of the result for general subgraphs $H$.
> However, we note that from a theoretical perspective, this is still significantly better than previous results.
> Please see the response to W1 from Reviewer hrky for all the details.
>
> Q1. For the DP algorithms given in the paper, the output A(D) is an approximation of the quantity that we want to compute (e.g., the number of triangles). But the approximation ratio of A(D) is never mentioned, i.e. what is the multiplicative error and what is the additive error? Furthermore, how could you compare these algorithms to previous algorithms in terms of approximation ratios?
>
> *Response*. Thanks for the question.
> We will elaborate on this in the paper.
> For the triangle counting problem, the approximation $A(D)$ is obtained from the diamond sampling result of (Ballard et al., 2015), and their error terms are specified in Lemma 3 and Theorem 4 of their paper (these depend on on the \#samples).
> Theorem 2 in our paper gives the precise approximation ratio for triangle counting---this includes both the multiplicative and additive factors.
> We will discuss this better, to contrast the error terms from those in (Ballard et al., 2015).
> In our experiments, we show the exact error.
> We will also add a comparison to the diamond sampling method, as also mentioned in W2 by Reviewer tb5j.
> We will consider the Triangle Count Relative Error and the time complexity in our comparison.
> Further, as mentioned in the response to W3 of reviewer tb5j, we will also add a comparison with the ladder function based method of Zhang et al., 2015.
>
>
> Q2.What is the state-of-the-art of running times of the DP algorithms for counting subgraphs with ell edges?
>
> *Response*. As mentioned in the response to W1 from Reviewer hrky,
> precise bounds have been considered for very few subgraph problems.
> In particular, for $k$-stars, the best running time is $O(m\log{n})$, and the best running time for $k$-triangles is $O(m\max_v d(v))$.
> For general subgraphs, prior papers do specify any efficient algorithms, and direct implementations of the smooth sensitivity technique would take $O(n^{\log{n}})$ time.
> We will elaborate on this in the Related Work section of the final version.
>
>
>
> Q3.Why the experiments are only for triangles? What about paths or other subgraphs?
>
> *Response.* The results for paths and general subgraphs are presented from a theoretical perspective and the algorithms are quite involved. The results show how to use multi-level local sensitivity with approximation.
> They improve on the previous best known results, but are still quite impractical and more efficient algorithms are needed for empirical evaluation, as mentioned in the response to W1 from Reviewer hrky.
> We will clarify this in the final version.

---

> ### Author Response · Authors · 2023-08-17
>
> We request the reviewer to let us know if our responses are adequate and if any other points need clarification.

---

> > ### Comment · Reviewer_Bhh2 · 2023-08-18
> >
> > I appreciate your thorough response. Upon reviewing the submission once more, I've come to realize that the presentation of the main results may lead to confusion or misinterpretation. Notably, there appears to be an unclear delineation between the tradeoff involving the utility (or approximation error) and privacy of your algorithms. Furthermore, the depiction of the running times of your algorithms seems to be causing confusion. For instance, although you assert the existence of a quasilinear time algorithm for privately counting the number of triangles, your actual demonstration pertains solely to a quasilinear time algorithm for Smooth Sensitivity. This distinction needs to be clarified in your presentation.
> >
> > I am in agreement with the assessment made by reviewer tb5J, suggesting that a more detailed and comprehensive comparison with prior research is essential. This inclusion would undoubtedly provide a clearer context for understanding your contributions and differentiating them from existing works. I encourage you to address these aspects in order to enhance the clarity and impact of your submission.

---

> > > ### Author Response · Authors · 2023-08-21
> > >
> > > We thank the reviewer for the detailed comments and suggestions. Please see below for our responses, and request for some clarifications, which we can respond to.
> > >
> > > Q1. comprehensive comparison with prior research is essential
> > >
> > > $Response$. We thank the reviewer for the advice. As we mentioned in the response to reviewer LedL, we will improve our related work section, and will add a detailed comparison to prior works.
> > >
> > > Q2. Notably, there appears to be an unclear delineation between the tradeoff involving the utility (or approximation error) and privacy of your algorithms.
> > >
> > > $Response$. We request the reviewer for clarification on this point. We discuss only one tradeoff between utility and privacy  (Figure 3), following the common type of utility-privacy tradeoff in prior papers. We show a different analysis of the error in the smooth sensitivity computed by our algorithm, referred to as Sensitivity Approx. Error in lines 323-324, and shown in Figure 4.
> > > This is not a utility-privacy tradeoff.
> > >
> > > We will clarify this further in the final version, but if the reviewer can mention if there is anything specific in the delineation that is unclear, it would be very helpful for us.
> > >
> > > Q3. the running times of your algorithms seems to be causing confusion. For instance, although you assert the existence of a quasilinear time algorithm for privately counting the number of triangles, your actual demonstration pertains solely to a quasilinear time algorithm for Smooth Sensitivity.
> > >
> > > $Response$. We would request clarification on this point as well. We claim quasilinear time only for triangle counting (lines 55-56).  Our algorithm is actually based on approximate smooth sensitivity, which we demonstrate in section 6. We state the running times for our other algorithms in lines 59-68---these are polynomial time if the subgraph size $\ell$ is a constant, but we are not claiming quasilinear time. We would also like to note that Algorithm 2 outputs the approximate smooth sensitivity for triangle counting, and its performance is summarized in Theorem 2.
> > >
> > > It is likely we have misunderstood the point, and a clarification would be very helpful.

---

### Official Review · Reviewer_hrky · 2023-07-06

**Soundness:** 3 good
**Presentation:** 3 good
**Contribution:** 3 good
**Rating:** 6
**Confidence:** 3

**Summary:**

This paper extends the study of local sensitivity and smooth sensitivity for the problem of subgraph counting. Previous work has identified local sensitivity and its derivates like smooth sensitivity as a tool to calibrate noise to the given input instead of global sensitivity. However, this comes with two challenges: it is often not easy to get a good bound on local sensitivity in a computationally efficient way, and local sensitivity itself can exploit privacy (which lead to the notion of smoothed sensitivity, a DP version of local sensitivity). In this submission, the authors study how an approximation of the smooth sensitivity and higher-order local sensitivity can be used privately to release an approximation of subgraphs counts, in particular triangles, paths and arbitrary constant-size graphs.

For triangle count, they show that one can release an approximation in $O(polylog(m+n))$ time plus the time needed to approximate the triangle count non-privately, by leveraging a fast estimation scheme for the smooth sensitivity and a diamond sampling technique from non-private algorithms. For general graphs, they extend the existing first and second order analysis of local sensitiviy for triangles and cliques to the $\ell$th order for graphs $H$ with $\ell$ edges. They derive that one can compute an approximation of the $\ell$th order local sensitivity and use it to release the count of copies of $H$ in the input graph in $O(n^{2\ell})$ time for arbitrary $G$ and in time $O(n^{\ell+1})$ if $H$ is tree. In an experiment section, they show that for several datasets of size xxK to xxxK nodes, their DP approximation algorithm for counting triangles provides one to three order of magnitude speedup, while the approximation is significantly less than a magnitude worse compared to the baseline based on exact smooth sensitivity.

Overall, this paper is a nice step into a directon that I think is promising to overcome the issue with global sensitivity on graphs. The theoretical bounds on running time are not practical, but they generalize from specific instances or classes of graphs to arbitrary graphs. This is often a first step into understanding a problem better. The experiments show that the approach can also improve the state of the art where the problem is already better understood in general, i.e., in the case of triangles.

**Strengths:**

* The paper provides general utility bounds for counting arbitrary subgraphs that are much more tailored to the actual input than approaches based on global sensitivity.
* The experiments indicate that algorithms also performs better in practice.

**Weaknesses:**

* The bound for general graphs and paths is impractical, as the degree of the running time polynomial scales linearly with the size of $H$.

**Questions:**

-

**Limitations:**

-

---

> ### Author Rebuttal · Authors · 2023-08-09
>
> W1. The bound for general graphs and paths is impractical, as the degree of the running time polynomial scales linearly with the size of $H$.
>
> *Response.* We completely agree with the reviewer's assessment about the practicality of the result for general subgraphs $H$.
> However, we note that from a theoretical perspective, this is still significantly better than previous results.
> For instance, a simple implementation of the smooth sensitivity result of (Nissim et al., 2007) and (Karwa et al., 2014) for counting subgraphs could take $O(n^{\log{n}})$ time for constant size subgraphs $H$, in the worst case, and they show efficient results only for two classes of subgraphs.
> Similarly, the technique of (Zhang et al., 2015), is potentially relevant for any subgraphs, but no efficient algorithms are presented for general subgraphs.
>
> Therefore, our results for general graphs and paths are more from a theoretical perspective.
> They demonstrate the power of using multi-level local sensitivity when combined with approximate queries and these are the first such results.

---

> > ### Comment · Reviewer_hrky · 2023-08-14
> >
> > Thanks for your response! I'll stay with my rating for now, but will reevaluate once there is a conclusion in the discussion with reviewer LedL.

---

> > > ### Author Response · Authors · 2023-08-17
> > >
> > > Please see our response to reviewer LedL.

---

### Official Review · Reviewer_tb5J · 2023-07-07

**Soundness:** 3 good
**Presentation:** 3 good
**Contribution:** 2 fair
**Rating:** 4
**Confidence:** 5

**Summary:**

This paper provides an algorithm to approximately estimate the smooth sensitivity for counting triangles in a graph, while preserving edge-level differential privacy. Compared with the existing solution for smooth sensitivity, the proposed algorithm demonstrates efficiency improvement in terms of computation.


**Strengths:**

1. Efficiently computing/estimating the smooth sensitivity for subgraph counting is a well-motivated problem.

2. The idea of the proposed algorithm is intuitive and the analysis seem to be sound.

**Weaknesses:**

1. It would be better if the authors can provide a table comparing their method with existing solutions. The comparison should include time complexities, tasks (e.g., triangle counting, clique counting), sample complexities, and maybe utility guarantees.

2. The original non-private and approximate algorithm for triangle counting (the diamond sampling [2]) and also the state-of-the-art solution(s) for approximate triangle counting without DP should be included for theoretical analysis and empirical evaluation. Including them would help the readers better understand the trade-offs between privacy and utility and between utility and efficiency.

3. Some baselines are missing in the empirical evaluation. For example, the inverse sensitivity, propose-test-release, and ladder functions.

4. The algorithms in Section 5 do not have empirical evaluations.

5. A more detailed discussion on the parallelism is needed. How much does parallelism account for the efficiency improvement of the proposed solution?

**Questions:**

See weaknesses.

**Limitations:**

Yes.

---

> ### Author Rebuttal · Authors · 2023-08-09
>
> W1.It would be better if the authors can provide a table comparing their method with existing solutions. The comparison should include time complexities, tasks (e.g., triangle counting, clique counting), sample complexities, and maybe utility guarantees.
>
> *Response:* Thanks for the suggestion.
> We will add such a table which will compare the time complexities for different subgraphs, such as triangles, $k$-stars and $k$-cliques, summarizing the references mentioned in our related work, and the recent survey of (Li et al., ACM Computing Surveys, 2023).
> We note that only the time complexity is mentioned in these papers, e.g., for triangle counting, our result in Theorem 2 is smaller than $O(m\cdot polylog(m))$, compared to $O(m\max_v d(v))$.
> Therefore, we will focus on the time complexity in the table.
>
>
> W2. The original non-private and approximate algorithm for triangle counting (the diamond sampling [2]) and also the state-of-the-art solution(s) for approximate triangle counting without DP should be included for theoretical analysis and empirical evaluation. Including them would help the readers better understand the trade-offs between privacy and utility and between utility and efficiency.
>
> *Response:*
> Our results in Section 6 show the error with respect to the $True Count_{\Delta}$---please see the metric {\em Triangle Count Relative Error} $=\frac{|Private Count_{\Delta} - True Count_{\Delta}|}{True Count_{\Delta}}$ and Figure 3 in Section 6.
> We will also show the error with respect to diamond sampling in the final version, as suggested.
> We note that since the diamond sampling paper would also incur some error due to sampling, it is quite likely that the Triangle Count Relative Error would reduce if we compare with the diamond sampling result, instead of $True Count_{\Delta}$.
>
>
> W3. Some baselines are missing in the empirical evaluation. For example, the inverse sensitivity, propose-test-release, and ladder functions.
>
> *Response.*
> Of the methods mentioned here, a result for triangle counting is known only using ladder functions (Zhang et al., 2015).
> We will add this method to the empirical evaluation section in the final paper.
>
> W4. The algorithms in Section 5 do not have empirical evaluations.
>
> *Response.*
> The algorithms in Section 5 are quite involved, and we present this more from a theoretical perspective.  They demonstrate the power of using multi-level local sensitivity when combined with approximate queries. These are the first such results.
> We expect more efficient algorithms in future work.
> We note that for most subgraph counting problems, the best theoretical bounds are much worse, and no practical evaluations are known for them.
> For instance, a simple implementation of the smooth sensitivity result of Nissim et al. for counting subgraphs could take $O(n^{\log{n}})$ time, but they show a better bound of $O(m\max_v d(v))$ for triangles; similarly, the ladder function technique of (Zhang et al., 2015) is potentially relevant for any subgraphs (if suitable ladder functions can be constructed), but they give practical algorithms only for $k$-stars and $k$-cliques.
>
> W5. A more detailed discussion on the parallelism is needed. How much does parallelism account for the efficiency improvement of the proposed solution?
>
> *Response.* As in the diamond sampling paper, our method can be easily parallelized. In short, the heaviest computing task is in Algorithm 4 (i.e., estimate $a_{ij}^2$) with high accuracy (i.e, with a large number of iterations s). Lines 5-10 of that algorithm can be easily parallelized to sample s values concurrently with a very slim chance of occuring race conditions.
> We will elaborate on this in the final version.

---

> ### Author Response · Authors · 2023-08-17
>
> We request the reviewer to let us know if our responses are adequate and if any other points need clarification.

---

> > ### Author Response · Authors · 2023-08-21
> >
> > Dear reviewer,
> > We are following up to see if you have any questions regarding our responses. We had done a comparison with two baselines. We had sent the anonymized link to the figures  to the AC per Neurips guideline. Please also see our response to reviewer LedL regarding comparison with related work. If you have are any additional questions, please let us know.

---

> > > ### Comment · Reviewer_hrky · 2023-08-21
> > >
> > > Dear Area Chair, Could you make the figures sent by the authors available?

---

### Decision · Program_Chairs · 2023-09-21

**Decision:**

Accept (poster)

**Comment:**

This paper studies private subgraph counting using approximation algorithms for the quantity and its sensitivity. The main contribution is a simple (in hindsight) way to use these approximation to release a private approximation. The authors show that this general approach leads to better algorithms for some common problems on graphs.
The paper received detailed reviews and an active discussion phase. The reviewers largely agreed that the paper is above bar for this conference and I am happy to recommend acceptance. I encourage the authors to incorporate all reviewer feedback in the final version of the paper.